# Nuclear elongation during spermiogenesis depends on physical linkage of nuclear pore complexes to bundled microtubules by *Drosophila* Mst27D

**Pengfei Li[ID], Giovanni Messina[ID]¤, Christian F. Lehner[ID]***

Department of Molecular Life Science (DMLS), University of Zurich, Zurich, Switzerland

¤ Current address: Dipartimento di Biologia e Biotecnologie "Charles Darwin", Sapienza Università di Roma, Rome, Italy

\* christian.lehner@imls.uzh.ch

**Data Availability Statement:** All relevant data are within the paper and its Supporting Information files.

## Abstract

Spermatozoa in animal species are usually highly elongated cells with a long motile tail attached to a head that contains the haploid genome in a compact and often elongated nucleus. In *Drosophila melanogaster*, the nucleus is compacted two hundred-fold in volume during spermiogenesis and re-modeled into a needle that is thirty-fold longer than its diameter. Nuclear elongation is preceded by a striking relocalization of nuclear pore complexes (NPCs). While NPCs are initially located throughout the nuclear envelope (NE) around the spherical nucleus of early round spermatids, they are later confined to one hemisphere. In the cytoplasm adjacent to this NPC-containing NE, the so-called dense complex with a strong bundle of microtubules is assembled. While this conspicuous proximity argued for functional significance of NPC-NE and microtubule bundle, experimental confirmation of their contributions to nuclear elongation has not yet been reported. Our functional characterization of the spermatid specific Mst27D protein now resolves this deficit. We demonstrate that Mst27D establishes physical linkage between NPC-NE and dense complex. The C-terminal region of Mst27D binds to the nuclear pore protein Nup358. The N-terminal CH domain of Mst27D, which is similar to that of EB1 family proteins, binds to microtubules. At high expression levels, Mst27D promotes bundling of microtubules in cultured cells. Microscopic analyses indicated co-localization of Mst27D with Nup358 and with the microtubule bundles of the dense complex. Time-lapse imaging revealed that nuclear elongation is accompanied by a progressive bundling of microtubules into a single elongated bundle. In *Mst27D* null mutants, this bundling process does not occur and nuclear elongation is abnormal. Thus, we propose that Mst27D permits normal nuclear elongation by promoting the attachment of the NPC-NE to the microtubules of the dense complex, as well as the progressive bundling of these microtubules.

**Funding:** The research was supported by funds obtained from the Swiss National Science Foundation (www.snf.ch), grant number 31003A_179433 (CFL). The funders had no role in study design, data collection and analysis, decision to publish, or preparation of the manuscript.

**Competing interests:** The authors have declared that no competing interests exist.

## Author summary

The sperm is a cell highly specialized for delivery of a haploid set of paternal chromosomes to the oocyte. Typically, it has a moving tail for swimming and a compact often elongated head containing specially packed chromosomes. During spermiogenesis in *Drosophila melanogaster* flies, for example, the nucleus is re-modeled from sphere into elongated needle with a two hundred-fold smaller volume. Nuclear elongation is preceded by a striking redistribution of nuclear pore complexes (NPCs) in the nuclear envelope (NE). At the start of nuclear elongation, NPCs are confined to a hemisphere of the spherical nucleus, and a strong bundle of microtubules is assembled in the cytoplasmic dense complex adjacent to the NPC-NE. Whether and how microtubule bundle and NPC-NE cooperate to achieve nuclear elongation has remained unresolved. Here, we report a functional characterization of Mst27D. This protein functions as a linker, connecting NPCs with microtubules that are bundled progressively into fewer and eventually into a single large bundle within the dense complex, according to our time-lapse imaging of nuclear elongation. In mutants lacking Mst27D, bundling does not proceed and nuclear elongation is defective. Thus, our findings advance the understanding of nuclear elongation in spermatids to the molecular level.

## Introduction

Sexual reproduction in animals entails fertilization of an oocyte by a spermatozoon. The initial description of the characteristic morphology of the spermatozoon by van Leeuwenhoek dates to the seventeenth century. Typically, animal spermatozoa including those of humans and *Drosophila melanogaster* have a compact head linked via a neck to a long motile tail. Their compact elongated shape is generally thought to reflect an adaptation for efficient swimming.

In humans and *D. melanogaster*, as well as in many other animal species, compaction of the sperm head, which contains the haploid genome, is achieved by a profound reorganization of DNA packaging [1–3]. In contrast to conventional nucleosomal chromatin, where DNA is wrapped around histone octamers, the genome in the sperm nucleus is compacted more tightly by sperm nuclear basic proteins (SNBPs). Histones are replaced in steps during spermiogenesis by transition proteins and eventually by protamines in mammals and by MST-HMG family proteins in *D. melanogaster* [4–9] (Fig 1A). Nuclear volume reduction during spermiogenesis is around 10- and 20-fold in humans and mouse, respectively, [10] and 200-fold in *D. melanogaster* [11,12] (Fig 1A and 1B).

Beyond volume reduction, transformation of nuclear shape is also extensive during spermiogenesis. While a spherical nucleus is present in early round spermatids, i.e., the haploid cells formed after completion of the second meiotic division (M II), a distinctive nuclear geometry is displayed in sperm. In human sperm, the nucleus is an ellipsoid with anterior flattening. In rodents, the sperm nucleus has a characteristic anterior hook, and in *D. melanogaster*, it is a highly elongated needle, 10 μm long and 0.3 μm in diameter [11,12] (Fig 1A and 1B). Clearly, the diversity in nuclear shape among animal sperm is remarkable, as already suggested by Leeuwenhoek's pioneering studies in a wide range of species. Despite conserved physiological roles and archetypical structural elements, sperm is a most rapidly evolving cell type, presumably reflecting sexual selection associated with sperm competition and additional drivers, including sexual conflict, reproductive isolation, germline pathogens, and mutations causing segregation distortion in the male germline [13–17]. The extreme compaction of the sperm nucleus into an elongated needle in *D. melanogaster*, for example, might not represent an

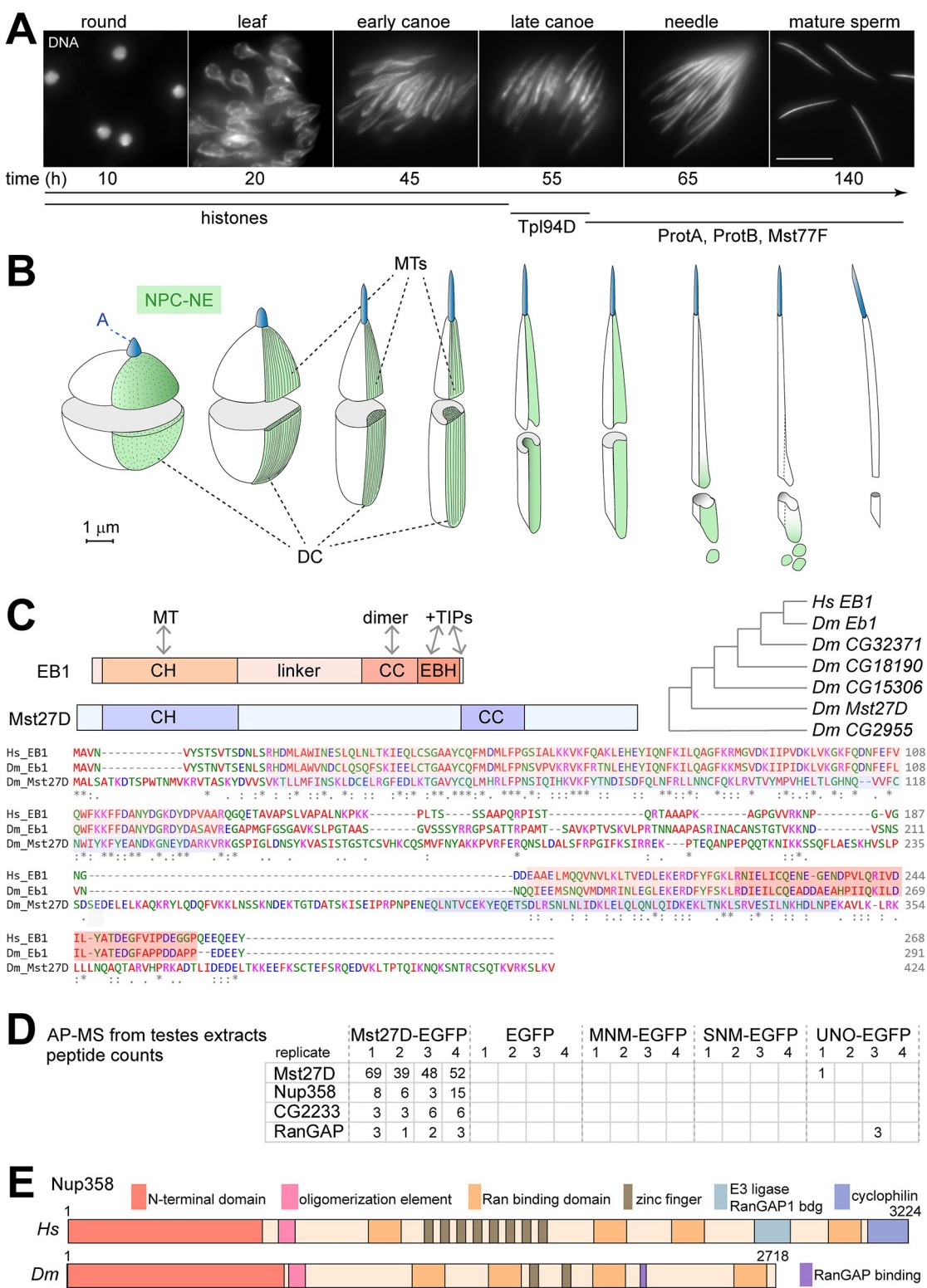

**Fig 1. Mst27D, a candidate factor important for nuclear elongation during spermiogenesis, associates with Nup358. (A,B)** Overview on nuclear remodeling during spermiogenesis in *D. melanogaster*. The spherical nucleus of early round spermatids is transformed into the compact elongated needle evident in sperm. (**A**) Stages of nuclear re-modeling are named according to the characteristic nuclear shape revealed by DNA staining of testis squash preparations. Scale bar = 10 μm. Estimates for time (hours) after completion of the second meiotic division are indicated below the images. Nuclear compaction is accompanied by a stepwise

replacement of histones with transition proteins like Tpl94D and later with sperm nuclear basic proteins including ProtA, ProtB and Mst77F. (**B**) Summary of characteristic features detected by ultrastructural analysis of nuclear remodeling (based on [12]). Nuclear pore complexes (NPCs) are first redistributed into a part of the nuclear envelope (NE) that covers only a hemisphere of the spherical nucleus in early spermatids. Microtubules (MTs) then assemble into a strong bundle in the cytoplasm next to the NPC-NE. This dense complex (DC) remains in tight association with the underlying NPC-NE during nuclear elongation. After completion of nuclear elongation, the MTs of the DC are disassembled and the NPC-NE is separated away from the elongated nucleus of late spermatids. The acrosome (A) is in front of the anterior end of the nucleus. (**C**) Mst27D and the EB1 protein family. EB1 binds to MTs with its N-terminal calponin homology domain (CH) and dimerization is mediated by the coiled coil region (CC). The EB1 homology domain (EBH) and the C-terminal EQ/DEEY motif recruit various proteins (+TIPs) to plus ends of growing MTs. *Mst27D* is relatively diverged among the EB1-related genes of *D. melanogaster* (*Dm*) as indicated by the cladogram representing amino acid sequence similarity detected in pairwise comparisons with EB1 of *Homo sapiens* (*Hs*). Similarity between Mst27D and EB1 of *Hs* and *Dm* is largely restricted to the CH domain. However, secondary predictions reveal a coiled coil region (CC) also in Mst27D. (**D**) Anti-EGFP beads were used for affinity-purification (AP) of the indicated target proteins from testes extracts, followed by identification of co-purified proteins by mass spectrometry (MS). Four replicate purifications were analyzed for each target. According to peptide counts, Nup358 was most efficiently and specifically co-purified with Mst27D-EGFP, as indicated by the list with the four proteins that were most abundant and specifically detected in the Mst27D-EGFP AP-MS samples. Empty cells indicate absence of detected peptides. See S1 Table for complete list. (**E**) Domain structure of Nup358 from *Hs* and *Dm*.

adaptation for efficient swimming, as muscle action during copulation already brings the sperm into the female sperm receptacle. Instead, the needle shape might reflect selective pressure to minimize the diameter of the micropyle, the pore in the eggshell through which sperm and occasionally perhaps harmful material can enter the oocyte. In humans compared to other mammals, variability in sperm and sperm head morphology is surprisingly high, even within individual males with normal fertility [18], and clinical benefits can be expected from an improved, molecular understanding of the processes that direct morphological differentiation [19].

Nuclear shape transformation during spermiogenesis in humans and *D. melanogaster* is accompanied by a striking relocalization of nuclear pore complexes (NPCs) as revealed initially by ultrastructural analyses [11,12,20–23]. NPCs, assembled from around 30 distinct nuclear pore proteins (Nups), permit nucleocytoplasmic exchange of material through the nuclear envelope (NE) composed of inner (INM) and outer (ONM) nuclear membrane (for a recent review see [24]). NPCs are usually distributed throughout the NE. While a uniform distribution is also observed initially in early round spermatids, mature human sperm have NPCs exclusively in a limited NE subregion above the basal body [25,26]. The manchette, a basket-like structure formed primarily from bundles of microtubules (MTs) around the mammalian spermatid nucleus during nuclear shape transformation has been implicated in NPC relocalization based on spatial correlations revealed by electron microscopy (EM) [23]. In *D. melanogaster*, NPCs are also uniformly distributed throughout the NE in germline cells of the testis initially. This distribution is observed in germline stem cells, in transit amplifying spermatogonial cells and in the resulting cysts with 16 interconnected spermatocytes that grow and mature over the stages S1 –S6 [27] before progressing through the meiotic divisions. Thereafter, the germline cells, now designated as spermatids, progress through spermiogenesis, which includes the drastic nuclear remodeling over characteristic stages (Fig 1A and 1B). Early after M II, in round spermatids, NPCs are still present throughout the NE of the spherical nuclei [11,12,20]. At a later stage, however, only a hemisphere of the NE contains NPCs, as first revealed by EM analyses [11,20] (Fig 1B). This redistribution of NPCs from spherical to hemispherical is completed before the onset of the nuclear shape transformation. After the NPC redistribution, conspicuous MT bundles are assembled along the NPC-containing NE (NPC-NE) in the so-called dense complex (DC), followed by a concerted elongation of nucleus and DC [11,20] (Fig 1B). After completion of nuclear elongation and compaction, the DC is disassembled and the NPC-NE is separated away from the nucleus in large vesicles for

elimination of the NE excess that is no longer needed to envelope the compacted needle-shaped nucleus [11,20] (Fig 1B). This shedding of the NPC-NE marks the start of the sperm individualization process that envelops each of the 64 spermatids, initially organized in a cyst with cytoplasmic interconnections, with a contiguous individual cell membrane [12,28,29].

Similar to the mammalian manchette [30,31], the DC in *D. melanogaster* spermatids has been suggested to be crucial for nuclear shape transformation [12,20,29]. In principle, elongation of the MT bundles in the DC might provide the force that promotes nuclear elongation. However, experimental support for this notion beyond EM analyses of fixed wild-type testis [11,20] is missing. Although the presence of a strong MT bundle within the DC during nuclear elongation is intriguing, it is not known whether and how the MTs in these bundles are physically linked to the NPC-NE, which is in close spatial association with the DC according to EM [11,20]. Clearly, a load-bearing connection between MT bundle of the DC and the NPC-NE is required if nuclear elongation is indeed enforced by the MT bundle.

Interestingly, a recent analysis of the *D. melanogaster* testis proteome revealed a candidate protein that may function in the suspected physical linkage between NPC-NE and MTs of the DC [32]. This candidate, Mst27D, might bind to MTs, as it shares sequence similarity with EB1 [32] (Fig 1C), a conserved protein well known to bind preferentially to growing plus ends of MTs (for reviews see [33,34]). Moreover, based on analyses with testes from males with an Mst27D-EGFP transgene, the subcellular localization of this protein during spermiogenesis appeared to be restricted to the region of NPC-NE and DC [32]. We report the results of our functional characterizations, demonstrating that Mst27D acts as a linker, which connects MTs of the DC with the Nup358 subunit of NPCs. Moreover, we show that Mst27D is crucial for the formation of the MT bundles of the DC and for normal nuclear elongation.

## Results

### Mst27D binds to Nup358

To identify interaction partners of Mst27D, we analyzed proteins that were co-purified with Mst27D-EGFP from testis extracts by mass spectrometry (MS). Mst27D-EGFP was expressed from a transgene (*g-Mst27D-EGFP*) under control of 5' *cis*-regulatory sequences derived from the *Mst27D* locus [32]. Around 9000 testes were isolated for each of the four replicate experiments from late larvae and early pupae using mass isolation [35]. Affinity-purification (AP) from testes extracts was achieved with anti-EGFP beads. To dismiss proteins that bind to GFP or to anti-GFP beads rather than to the Mst27D part of Mst27D-EGFP, control AP-MS experiments were done in parallel with testes expressing only EGFP. Moreover, data obtained in analogous AP-MS experiments with testes expressing MNM-EGFP, SNM-EGFP and UNO-EGFP [35] was used for comparison. Our results indicated that Nup358 was most efficiently co-purified specifically with Mst27D (Fig 1D and S1 Table). In addition, RanGAP, which is known to bind to Nup358 [36], was also among the proteins co-purified specifically with Mst27D (Fig 1D). Nup358 is the largest constituent of the metazoan NPC. Human Nup358 has 3224 amino acid (aa) residues. The *D. melanogaster* ortholog, which contains fewer zinc fingers and lacks E3 ligase and cyclophilin domains, still has 2718 aa residues (Fig 1E). The α-helical N-terminal domain (NTD) and the adjacent oligomerization element (OE) of human Nup358 were shown to mediate binding as homo-pentamers on the cytoplasmic periphery of the NPCs [37,38] (Fig 1E). The reminder of Nup358 extends as a major component of the cytoplasmic filaments of the NPC as far as 60 nm into the cytoplasm and includes four Ran-binding domains (RanBDs) [37] (Fig 1E). *D. melanogaster* Nup358 binds RanGAP with a 23-aa stretch that is present within Nup358-PB, the major isoform [36] (Fig 1E).

Apart from Nup358 and RanGAP, the product of *CG2233* appeared to be co-purified specifically with Mst27D-EGFP (Fig 1D). *CG2233* is an uncharacterized gene with homologs detectable only in dipterans. According to transcriptomic analyses, it is strongly expressed in several tissues and also in testes although at slightly lower levels [39]. Further characterization will be required to confirm a potential *CG2233* protein association with Mst27D. Beyond the proteins listed in Fig 1D, our AP-MS analyses identified some additional proteins that were co-purified with Mst27D-EGFP but not with control proteins (S1 Table). However, low numbers of detected peptides make future validation of these additional candidate interactors even more critical.

For confirmation of the interaction between Mst27D and Nup358, we performed co-immunoprecipitation experiments after transient transfection of cultured S2R+ cells with expression constructs. Mst27D was expressed with EGFP fused to the C terminus (Fig 2A) and Nup358 with mCherry at the N terminus (Fig 2B). Antibodies against mCherry allowed successful immunoprecipitation of mCherry-Nup358 (Fig 2C). Immunoblotting with anti-EGFP demonstrated that Mst27D-EGFP was co-immunoprecipitated efficiently with mCherry-Nup358 (Fig 2C).

Co-immunoprecipitation experiments with Mst27D protein fragments fused to EGFP (Fig 2A) demonstrated that the C-terminal (CT) part of Mst27D (aa 151–424) binds to Nup358 (Fig 2C). In contrast to Mst27D_CT-EGFP, Mst27D_CH-EGFP, i.e., the N-terminal Calponin homology (CH) domain (aa 1–150) fused to EGFP, was not co-immunoprecipitated with mCherry-Nup358 (Fig 2C).

To identify the region within Nup358 that binds to Mst27D, an initial series of N- and C-terminal truncations was generated (Fig 2B; N1-N5 and C1-C5). The truncated Nup358 fragments with N-terminal mCherry were co-expressed with Mst27D_CT-EGFP in S2R+ cells. After immunoprecipitation of the Nup358 fragments with anti-mCherry, immunoblotting with anti-EGFP indicated that it is the C-terminal region of Nup358 that binds to Mst27D_CT-EGFP (Fig 2D). Deletion of the C-terminal most region (aa 2342–2718) abolished co-immunoprecipitation of Mst27D_CT-EGFP (Fig 2D). Conversely, this most C-terminal Nup358 region was co-immunoprecipitated with Mst27D_CT-EGFP, albeit with lower efficiency compared to larger C-terminal fragments (Fig 2D). Microscopic analysis of the subcellular localization of the mCherry-Nup358 fragments indicated that NE localization was dependent on the presence of NTD and predicted OE (S1 Fig), as shown previously for *Hs* Nup358 [37,38]. Mst27D-EGFP also co-immunoprecipitated mCherry-Nup358 fragments that were not localized at the NE, suggesting that the interaction between Mst27D and Nup358 does not depend on other Nups and is thus likely direct. Further dissection with additional C-terminal truncations (Fig 2B; N6-N8) confirmed the importance of the C-terminal region of Nup358 for Mst27D binding. The additional truncations identified sequences C-terminal to aa 2538 as indispensable for Mst27D_CT-EGFP binding (Fig 2E, left). With full length Mst27D-EGFP, we obtained analogous results as with Mst27D_CT-EGFP (Fig 2E, right).

Overall, the results of our AP-MS analysis with extracts from *g-Mst27D-EGFP* testes and the co-immunoprecipitation experiments with S2R+ cells indicated that Mst27D binds to Nup358. This interaction is mediated by the C-terminal regions of these proteins.

## Mst27D dimers can localize to the NE and on MTs

Beyond binding to NPCs via Nup358, Mst27D is predicted to associate with MTs based on the aa sequence similarity of its CH domain with that of Eb1. To assess these predictions, we analyzed the localization of Mst27D-EGFP after co-expression with mCherry-Nup358 in S2R+ cells. Indeed, as expected, Mst27D-EGFP was observed to be enriched at the NE and on MTs

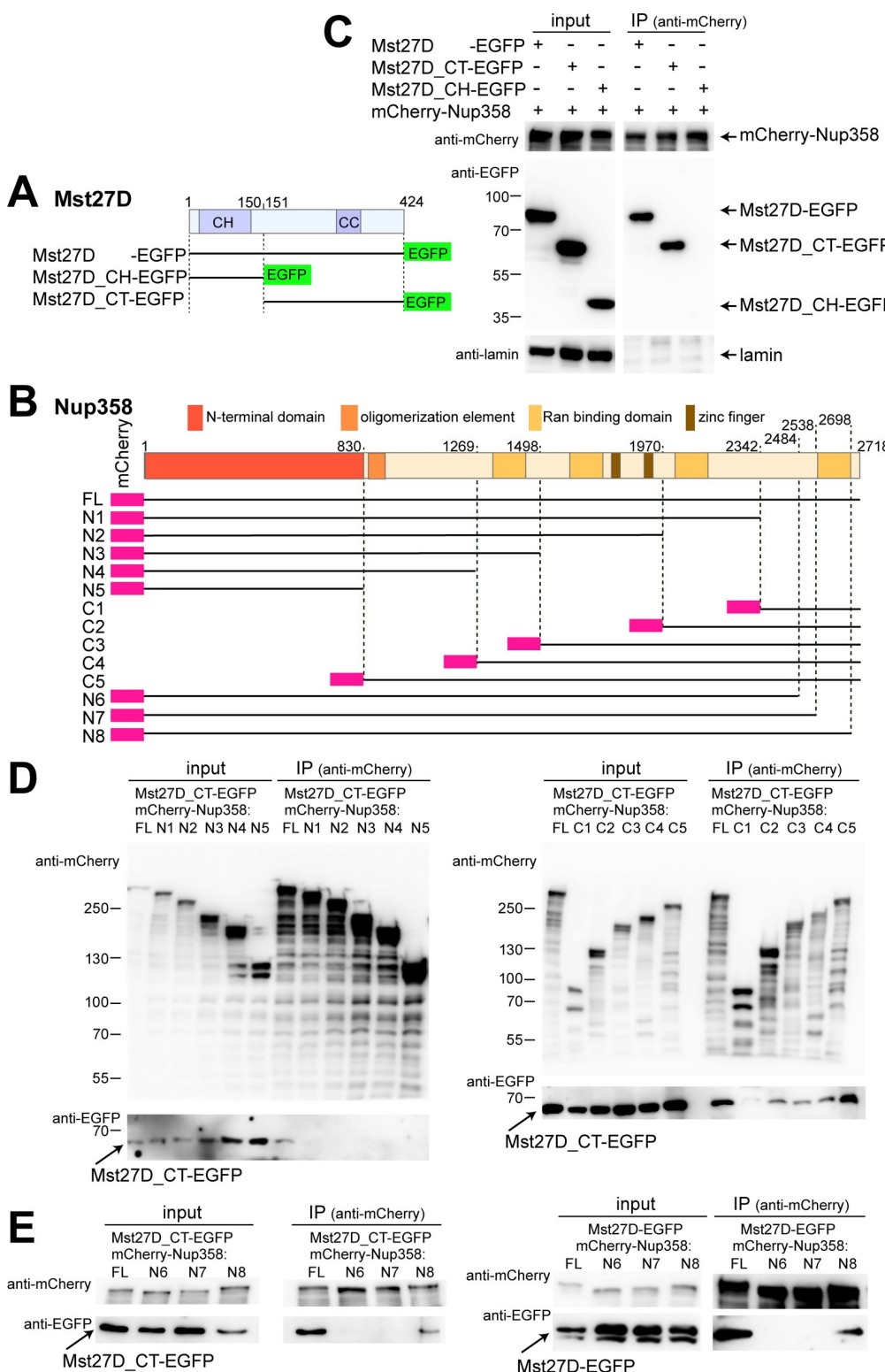

**Fig 2. Interaction between Mst27D and Nup358 is mediated by their C-terminal regions.** (**A-E**) Identification of interacting regions in Mst27D and Nup358 by co-immunoprecipitation experiments after transient co-expression in S2R+ cells. (**A,B**) Schemes illustrating the structure of the analyzed fragments of Mst27D (A) and Nup358 (B). (**C-E**) Co-immunoprecipitation after transient co-expression of the indicated fusion proteins. Presence or absence of proteins in the extracts used for immunoprecipitation (input) or in the samples immunoprecipitated with anti-

mCherry (IP) were analyzed by immunoblotting with anti-mCherry, anti-EGFP and anti-lamin. Positions of molecular weight markers are indicated.

(Fig 3A). Mst27D_CT-EGFP, encompassing the C terminal region but not the N-terminal CH domain of Mst27D, exhibited a stronger enrichment at the NE and a dissimilar association with MTs, compared to full length Mst27D-EGFP (Fig 3A). Mst27D_CT-EGFP appeared to highlight comets, comparable to growing MT plus ends visualized by Eb1 [40], while full length Mst27D-EGFP associated with MTs all along their length (Fig 3A and 3C). Somewhat unexpectedly, Mst27D_CH-EGFP, i.e., the N-terminal CH domain tagged with EGFP, did not associate with MTs (Fig 3A and 3C).

Coiled coil-mediated homodimerization stimulates MT binding and tracking of growing MT plus ends strongly in case of EB1 family proteins [41–45]. Unlike full length *D. melanogaster* Eb1, its N-terminal CH domain failed to bind to MTs except after fusion to a leucine zipper dimerization motif [43]. Therefore, MT binding by the CH domain of Mst27D might require dimerization. While aa sequence similarity between *D. melanogaster* Eb1 and Mst27D is very low outside the N-terminal CH domain, secondary structure predictions indicated a region with coiled-coil forming propensity within the C-terminal part of Mst27D (Fig 1C). To assess whether Mst27D might dimerize, we performed co-immunoprecipitation experiments after co-expression of Mst27D-mCherry and Mst27D-EGFP in S2R+ cells. In parallel, Mst27D-mCherry was also co-expressed with either Mst27D_CH-EGFP, Mst27D_CT-EGFP or *D. melanogaster* Eb1-EGFP. Our results indicated that Mst27D-mCherry co-immunoprecipitated Mst27D-EGFP and Mst27D_CT-EGFP, but neither Mst27D_CH-EGFP nor Eb1-EGFP (Fig 3B). Thus, Mst27D appears to form homodimers mediated by the CT region.

To address whether a homodimerizing version of the CH domain of Mst27D might bind to MTs, we expressed an EGFP-tagged chimeric Mst27D-Eb1 protein in S2R+ cells. The N-terminal domain in this chimera was the CH domain of Mst27D and the C-terminal region was that of *D. melanogaster* Eb1 (Fig 3C). The subcellular localization of Mst27D_CH-Eb1_CT-EGFP was analyzed by time-lapse imaging of live cells stably expressing the chimeric protein. For comparison, additional S2R+ cell lines expressing either full length Mst27D-EGFP, Mst27D_CH-EGFP, Mst27D_CT-EGFP or Eb1-EGFP were analyzed analogously (Fig 3C). The chimeric Mst27D_CH-Eb1_CT-EGFP protein was strongly associated with MTs during both interphase and mitosis (Fig 3C). In contrast, Mst27D_CH-EGFP without the Eb1_CT region was not MT associated (Fig 3C), as previously observed with fixed preparations (Fig 3A). The localization of the chimeric Mst27D_CH-Eb1_CT-EGFP protein on MTs was similar to that of full length Mst27D-EGFP (Fig 3C) and clearly distinct from that of Eb1-EGFP (Fig 3C). While Eb1-EGFP highlighted moving comets, as expected [43,46], Mst27D-EGFP and the chimeric Mst27D_CH-Eb1_CT-EGFP were all along the MTs (Fig 3C).

Overall, these results demonstrate that the CH domain of Mst27D mediates binding to MTs and suggest that efficient MT binding depends on Mst27D homodimerization. Moreover, Mst27D binds all along MTs, contrasting with Eb1's preference for growing MT plus ends.

Time-lapse imaging of S2R+ cells expressing Mst27D_CT-EGFP (Fig 3C) clearly confirmed an enrichment on the NE. However, in addition, EGFP signals were also present on moving comets, weaker but highly similar to Eb1-EGFP (Fig 3C). Although heterodimerization of Mst27D-mCherry and Eb1-EGFP was not observed in our co-immunoprecipitation experiments (Fig 3B), we speculate that heterodimerization of Mst27D_CT-EGFP with endogenously expressed Eb1 or perhaps with the uncharacterized Eb1 family protein encoded by *CG18190*, which appears to be expressed in S2R+ cells [47], might explain the comet localization.

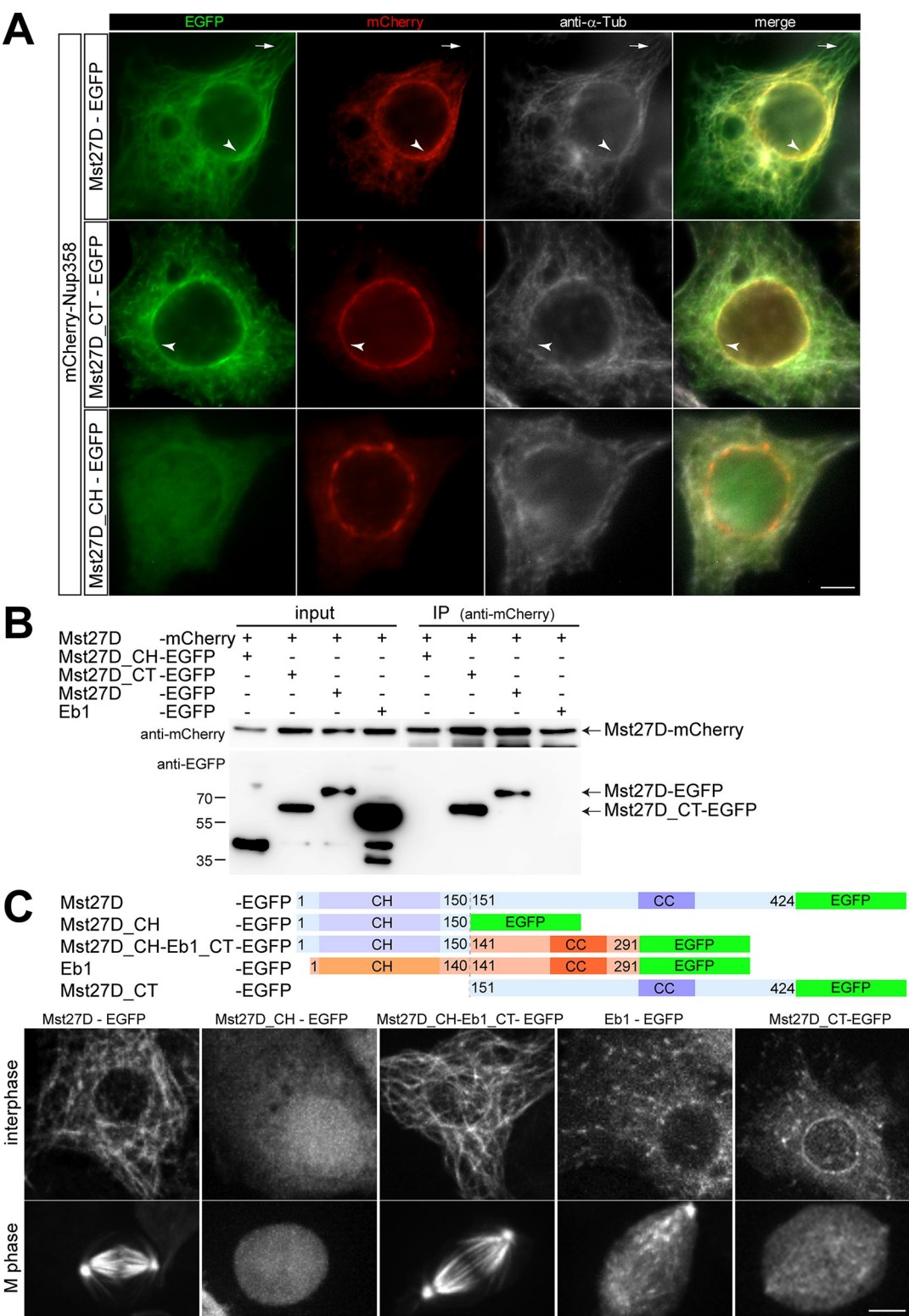

**Fig 3. Subcellular localization of Mst27D in S2R+ cells at the NE and on MTs.** (**A**) S2R+ cells expressing mCherry-Nup358 and either Mst27D-EGFP, Mst27D_CT-EGFP or Mst27D_CH-EGFP were fixed and labeled with anti-α-tubulin. Some distinctive EGFP signals on MTs (arrows) and at the NE (arrowheads) are pointed out. (**B**) Evidence for Mst27D dimerization. Mst27D-mCherry was co-expressed in S2R+ cells with EGFP fusion proteins as indicated. The presence or absence of proteins in the extracts used for immunoprecipitation (input) or in the material immunoprecipitated with anti-mCherry (IP) was

analyzed by immunoblotting with anti-mCherry and anti-EGFP. Proteins co-immunoprecipitated with Mst27D-mCherry are indicated (arrows), as well as the positions of molecular weight markers. (**C**) Subcellular localization of the indicated EGFP fusion proteins was analyzed by live imaging with S2R+ cell lines. Still frames display representative cells during interphase and mitosis, respectively. Scale bars = 5 μm.

The experiments involving expression of Mst27D-EGFP in S2R+ cells also provided evidence suggesting that Mst27D might have MT bundling activity. In cells with high expression levels, Mst27D-EGFP was localized in extended strong cables that were also intensely labeled with the tubulin stain SPY555-tubulin (S2 Fig). Intracellular cables were also present in S2R+ cells expressing Mst27D-mCherry at high levels (S2 Fig), indicating that MT bundling by Mst27D does not depend on the presence of EGFP, which has a weak propensity to form dimers, while mCherry is essentially monomeric [48]. Intracellular MT cables with Mst27D_CH-Eb1_CT-EGFP were also present in cells expressing high levels of this chimeric MT binding protein (S2 Fig). In contrast, cells expressing high levels of Mst27D_CH-EGFP, Mst27_CT-EGFP or *D. melanogaster* Eb1-EGFP did not display such cables.

Interestingly, the MT cables in cells with high levels of Mst27D-EGFP also contained mCherry-Nup358, if this protein was co-expressed (S3 Fig). In contrast, mCherry-Nup358 was not recruited into the cables induced by Mst27D_CH-Eb1_CT-EGFP (S3 Fig). These findings provide further support for the conclusion that Nup358 binds to the CT region of Mst27D, which is present in Mst27D-EGFP but not in Mst27D_CH-Eb1_CT-EGFP. Further support for the interaction between Nup358 and the CT region of Mst27D was obtained from an analysis of the effect of Nup358 depletion by RNAi on the localization of Mst27D_CT-EGFP in S2R+ cells (S3 Fig). Nup358 depletion abolished the localization of Mst27D_CT-EGFP at the NE (S3 Fig).

## Mst27D association with the NE during spermatogenesis depends on Nup358

To analyze the subcellular localization of Mst27D in the organism, we generated a fly line expressing Mst27D-mCherry and EGFP-Nup358 from transgenes. Expression of these transgenes, *g-Mst27D-mCherry* and *g-EGFP-Nup358*, was controlled by *cis*-regulatory sequences derived from the endogenous gene loci. As described in further detail below, the transgenes rescued *Mst27D* and *Nup358* null mutants, respectively, indicating that the tagged protein products were functional. While *Nup358* appears to be expressed ubiquitously, *Mst27D* expression is testis-specific [32,39]. Whole mount preparations of testes from males expressing *g-Mst27D-mCherry* and *g-EGFP-Nup358* revealed an onset of Mst27D-mCherry accumulation in spermatocytes at the S5 stage (Fig 4A). Mst27D-mCherry was clearly enriched at the NE. High magnification revealed precise co-localization of Mst27D-mCherry and EGFP-Nup358 at the NE (Fig 4A). Mst27D-mCherry and EGFP-Nup358 were present in elongated patches on the NE. Thus, NPCs appear to be clustered within the NE of late spermatocytes, in contrast to the even distribution of NPCs in the NE of cultured mammalian cells [49]. In *D. melanogaster,* however, super-resolution microscopy has recently revealed pronounced NPC clustering also in various additional tissues [50]. In contrast to EGFP-Nup358, which localized primarily at the NE, Mst27D-mCherry was also present in the cytoplasm (Fig 4A). The cytoplasmic Mst27D-mCherry signals were diffuse. Whether MT collapse during fixation contributed to the diffuse appearance of the cytoplasmic signals remains unresolved.

To assess the dynamics of the association of Mst27D with the NE *in vivo*, we generated a *g-Mst27D-Dendra2* transgenic fly line. After locally restricted photoconversion in a small region of the NE in S6 spermatocytes, the converted red fluorescent Mst27D-Dendra2 was detected

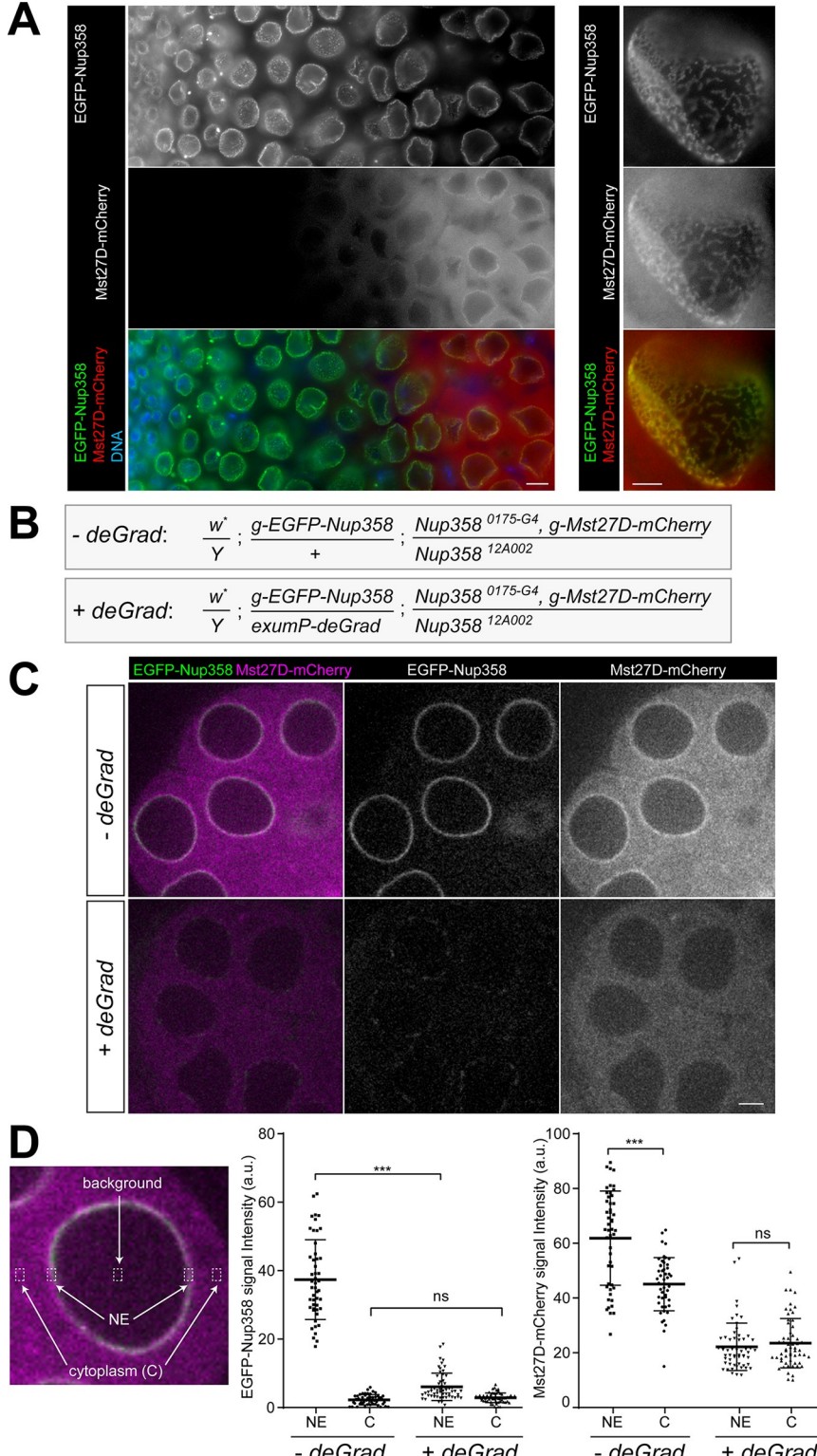

**Fig 4. Nup358 is required for NE localization of Mst27D-mCherry in late spermatocytes.** (**A**) Co-localization of Mst27D-mCherry after onset of expression in S5 spermatocytes with EGFP-Nup358. Whole mount preparations of testes from males with *g-Mst27D-mCherry* and *g-EGFP-Nup358* were fixed and labeled with a DNA stain. A maximum intensity projections of optical sections through the most peripheral germline cells in the testis tube are displayed in the left panel, with early stages in the apical testis region on the left and late spermatocytes on the right side. Single

optical sections grazing the NE are displayed at high magnification in the right panel. (**B**-**D**) The deGradFP system [52] was used for analysis of Mst27D-mCherry localization after EGFP-Nup358 degradation in *Nup358* mutant spermatocytes. (**B**) The indicated genotypes, *-deGrad* and *+deGrad*, were used for comparative analyses. (**C**) After live imaging of cysts released from early pupal testes, S6 spermatocytes were selected for quantitative analysis. Single optical sections through representative S6 spermatocytes are displayed. (**D**) Signal intensities of EGFP-Nup358 and Mst27D-mCherry at the NE and in the cytoplasm were quantified in regions of interests (ROIs) as indicated. For each cell, the intensities in the two cytoplasmic ROIs were averaged, as well as those in the two ROIs on the NE. The intensities observed within the ROI in center of the nucleus were considered to reflect background, which was subtracted from the intensities detected in the cytoplasm and at the NE. Swarm plots display signal intensities detected in individual cells as well as their mean (+/- s.d.); n = 49 from 9 distinct cysts (*-deGrad*) and 57 from 5 distinct cysts (*+deGrad*). Scale bars = 10 μm (A, left), 5 μm (A, right) and 10 μm (C).

within minutes throughout the NE (S4 Fig). In contrast, at least in cultured mammalian cells, where lateral NPC mobility within the NE has been analyzed, NPCs are essentially immobile [51]. Quantification of the dispersal of converted Mst27D-Dendra2 suggested that Mst27D in the cytoplasmic pool diffuses rapidly with intermittent episodes of transient association with and dissociation from the NE (S4 Fig).

To determine whether the association of Mst27D with NPCs is mediated by Nup358, we applied deGradFP [52]. This method permits elimination of GFP fusion proteins by expression of Nslmb-vhh4-GFP, an anti-GFP nanobody fused to the F-box of Slmb that is assembled into a GFP-specific ubiquitin ligase [52]. Hence, an analysis of Mst27D-mCherry localization after EGFP-Nup358 degradation in spermatocytes lacking endogenous Nup358 was envisaged. However, the original deGradFP system involves expression of Nslmb-vhh4-GFP with the *UAS/GAL4* system which does not promote efficient expression in spermatocytes during mid- to late stages. Therefore, alternative expression of Nslmb-vhh4-GFP using the transgenes *beta-Tub85DP-Nslmb-vhh4-GFP* and *exumP-Nslmb-vhh4-GFP* transgenes was explored (S6 Fig). These transgenes exploit spermatocyte-specific *cis*-regulatory sequences derived from the gene loci *betaTub85D* and *exuperantia (exu)*, respectively [53,54]. The transgene under control of *exumP* (*exumP-deGrad* in the following) resulted in earlier degradation during spermatocyte maturation and worked better for EGFP-Nup358 elimination just before the onset of *g-Mst27D-mCherry* expression (S5 Fig). The two genotypes that were compared for our analysis of the effects of EGFP-Nup358 elimination on Mst27D-mCherry (Fig 4B) will be designated as "*-deGrad*" and "*+deGrad*". The *exumP-deGrad* transgene was present only in the latter, but otherwise these two genotypes were identical. For analysis, we used testes dissected at the early pupal stages. As expected, EGFP-Nup358 signals were strongly reduced in late *+deGrad* spermatocytes, compared to the *-deGrad* control spermatocytes (S5 Fig). Presumably reflecting some unanticipated low-level expression of *exumP-deGrad* even at earlier stages of spermatogenesis, EGFP-Nup358 was also slightly weaker in *+deGrad* compared to *-deGrad* in the anterior of pupal testes where cells at early stages reside (S5 Fig).

For careful quantification of the effects of EGFP-Nup358 degradation on Mst27D-mCherry localization, spermatocyte cysts were released from pupal testes for live imaging with a spinning disc confocal microscope. We focused on cysts at the S6 stage, which has a short duration and can be identified readily due to the spherical shape of the nuclei. Signal intensities in the EGFP and mCherry channels at the NE and in the cytoplasm were quantified (Fig 4C and 4D). EGFP-Nup358 signals at the NE were almost completely abolished in *+deGrad* spermatocytes (Fig 4C and 4D). Moreover, the Mst27D-mCherry signals at the NE, which were clearly above the cytoplasmic level in *-deGrad* spermatocytes, were no longer above the cytoplasmic level in *+deGrad* spermatocytes (Fig 4C and 4D), indicating that Mst27D-mCherry was either no longer localized at the NE or at least strongly reduced. These results indicate that Nup358 is required for normal localization of Mst27D to the NE. Beyond the effect on localization of

Mst27D-mCherry, Nup358-EGFP degradation was also accompanied by an apparent reduction of the total level of Mst27D-mCherry (Fig 4C and 4D).

## Co-localization of Mst27D and Nup358 in spermatids during nuclear elongation

To compare the localization of Mst27D and Nup358 throughout spermatogenesis, we studied testes of males with the *g-Mst27D-mCherry* and *g-EGFP-Nup358* transgenes with fixed samples and by live imaging. In addition, we also analyzed an emerald-GFP (emGFP) knock-in allele of *Nup358* [55] combined with *g-Mst27D-mCherry*, which gave indistinguishable results.

During the meiotic divisions, nuclear envelope breakdown (NEBD) is incomplete in *D. melanogaster* spermatocytes [11]. Similar as also in other tissues and developmental stages [56–59], NPCs are disassembled during M I and M II, but a fenestrated NE encloses spindles and chromosomes almost completely during the meiotic divisions except for more prominent openings beneath the centrosomes at opposite spindle poles. Interestingly, EGFP-Nup358 was present on the spindle envelope during the meiotic divisions, while Nup58-EGFP was diffusely distributed throughout the cell (S6 Fig). In contrast, Mst27D-mCherry was primarily enriched on the spindle (S6 Fig). After completion of M II, EGFP-Nup358 first adopted an essentially symmetric distribution throughout the NE around the spherical nuclei of early round spermatids. However, within about two hours, the symmetric EGFP-Nup358 localization was transformed into a polarized asymmetric distribution where EGFP-Nup358 was restricted to a hemisphere of the NE (S6 Fig and S1 Movie). During this NE polarization process, Mst27D-mCherry was strongly enriched at the EGFP-Nup358-containing region of the NE (S6 Fig and S1 Movie).

At the start of nuclear elongation in spermatids, Mst27D-mCherry and EGFP-Nup358 were co-localized on the NE in a hemispherical cap. The two proteins remained co-localized during nuclear elongation that was accompanied by the spatial transformation of the hemispherical cap into a stripe running in a grove extended all along the elongated spermatid nucleus (Fig 5A). After completion of nuclear elongation, Mst27D-mCherry and EGFP-Nup358 lost their association with spermatid nuclei. The elimination of these two proteins from the elongated spermatid nucleus occurred in a process designated here as NPC-NE shedding. This process, which could also be monitored in *g-Nup58-EGFP* testes, occurred around the time when the F actin-containing individualization cones started to form (S7 Fig). Time-lapse imaging of spermatid cysts expressing *g-Nup58-EGFP* and a *g-ProtB-DsRed* transgene indicated that NPC-NE shedding was completed within approximately one hour soon after the onset of nuclear accumulation of ProtB-DsRed (S7 Fig and S2 Movie). The NPC-NE was observed to move in basal direction and was detached eventually from the spermatid nucleus as a single large vesicle (S7 Fig and S2 Movie). Thereafter, this vesicle was displaced along the axoneme into the waste bag at the end of the sperm tail. After NPC-NE shedding, Nup58-EGFP was absent from spermatid nuclei except for a very weak dot signal at the basal end of the elongated nucleus that remained detectable also in mature sperm (S7 Fig). During NPC-NE shedding, EGFP-Nup358 and Mst27D-mCherry were also co-localized (Fig 5A). However, after completion of NPC-NE shedding, Mst27D-mCherry signals disappeared rapidly rather than moving along with the NPC-NE vesicles into the waste bag (Fig 5A).

For verification of expression pattern and subcellular localization revealed by the *g-Mst27D-mCherry* transgene (or identically with *g-Mst27D-EGFP*), we generated antibodies against three distinct Mst27D peptides (Fig 5B). The antibodies readily detected Mst27D-EGFP expressed in S2R+ cells, except for those directed against a C terminal peptide (Fig 5C). Immunostaining of testes with all our anti-Mst27D antibodies resulted in

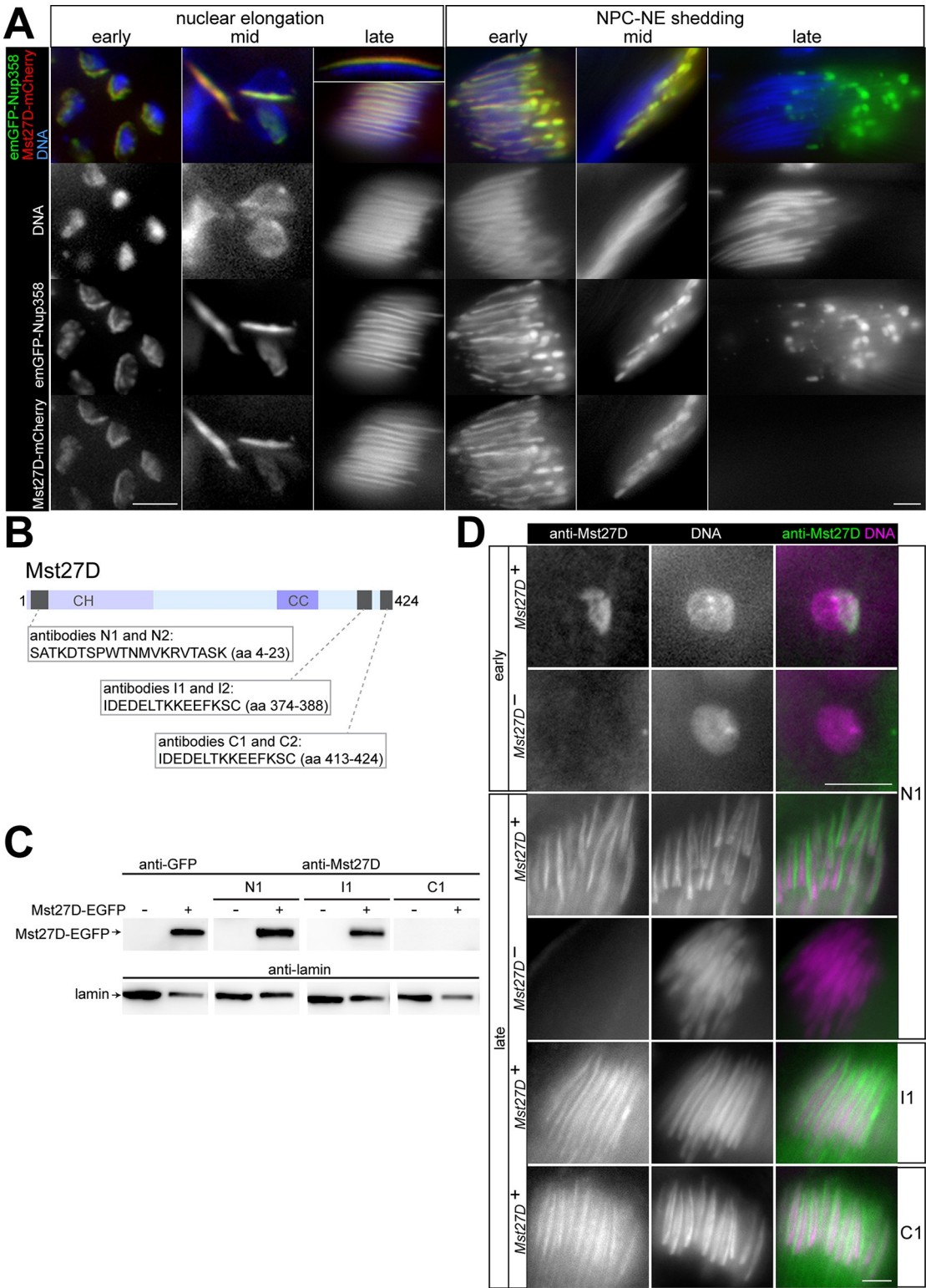

**Fig 5. Localization of Mst27D and Nup358 in spermatids during nuclear elongation and NPC-NE shedding.** (**A**) Testes from males with *g-Mst27D-mCherry* and *g-emGFP-Nup358* were fixed and labeled with a DNA stain. Regions from cysts during nuclear elongation and NPC-NE shedding are displayed. An isolated fully elongated nucleus is presented in the inset on top of the third column. (**B**-**D**) Antibodies against Mst27D. (**B**) The indicated peptides were used for production of affinity-purified polyclonal rabbit antibodies. (**C**) Total extracts from parental S2R+ cells (-) or from S2R+ cells expressing Mst27D-EGFP (+) were probed by

immunoblotting and the indicated antibodies. Re-probing with anti-lamin was done for comparison of loading. (**D**) Immunofluorescent labeling of whole mount preparations of testes with the indicated antibodies against Mst27D and a DNA stain. Testes were isolated from either control (*Mst27D*+) or *Mst27D* null mutant males (*Mst27D*-). Single optical sections displaying spermatids at the onset of nuclear elongation (early) and at the canoe stage (late) are displayed. Scale bars = 5 μm; while the bar on the left in (A) applies to the first two columns and the inset, the bar on the right applies to all other images.

comparable specific signals, i.e., those not observed in *Mst27D* null mutant testes (Fig 5D; see also below), indicating that the antibodies directed against the C terminal peptide recognize wild-type Mst27D but not Mst27D-EGFP.

The specific anti-Mst27D signals observed in testes confirmed the findings made with the transgenes although not completely. The fluorescently tagged Mst27D proteins expressed from the transgenes appeared to accumulate much earlier than endogenous Mst27D protein detected by anti-Mst27D. While the transgene products were already detectable in S5 spermatocytes (Fig 4A), specific anti-Mst27D signals emerged at the onset of nuclear elongation (Fig 5D).

In principle, the later onset of Mst27D accumulation revealed by anti-Mst27D staining might reflect reduced detection sensitivity compared with fluorescence of Mst27D fusion proteins expressed by the transgenes. Alternatively, the *cis*-regulatory sequences in the transgenes might direct an expression pattern distinct from that of the endogenous *Mst27D* gene. In *g-Mst27D-EGFP* [32] and analogously in our *g-Mst27D-mCherry* transgene, SV40 terminator sequences were used instead of the downstream sequences present at the endogenous *Mst27D* locus. For clarification of the significance of the downstream sequences, we generated a *g-Mst27D-EGFP-endo3'* transgene, in which the endogenous *Mst27D* downstream sequences were present rather than SV40 terminator sequences. Like anti-Mst27D signals, Mst27D-EGFP derived from *g-Mst27D-EGFP-endo3'* was not detectable before the onset of nuclear elongation, but it was present during the canoe stage at levels even higher than those resulting with the SV40 terminator containing transgenes (S8 Fig).

In conclusion, based on the concurrent results obtained with anti-Mst27D and *g-Mst27D-EGFP-endo3'*, endogenous Mst27D accumulation starts after meiosis in early spermatids and increases during nuclear elongation before disappearing at the onset of spermatid individualization.

## Mst27D is required for normal male fertility and normal nuclear elongation in spermatids

To determine the function of Mst27D, we characterized the phenotype of *Mst27D* mutants. A first mutant allele, *Mst27D*<sup>LL01973</sup> (hereafter abbreviated as *Mst27D*<sup>LL</sup>) was generated in a large-scale effort for isolation of *piggyBac* transposon insertion lines [60]. We confirmed the presence of a *piggyBac* insertion within the *Mst27D* coding sequence in the *Mst27D*<sup>LL</sup> allele by polymerase chain reaction (PCR) and DNA sequencing (Fig 6A). The predicted protein product that the *Mst27D*<sup>LL</sup> allele could express in principle comprises the first 172 aa residues of Mst27D followed by seven extra aa residues (PRKIIIL) and a premature stop codon. However, the antibodies N1 and N2 directed against an N-terminal peptide did not produce specific signals in *Mst27D*<sup>LL</sup> mutant testes, possibly because of non-sense mediated decay of *Mst27D*<sup>LL</sup> transcripts. A second *Mst27D* allele (hereafter designated as *Mst27D*<sup>cc</sup>) was generated using CRISPR/Cas9. *Mst27D*<sup>cc</sup>, which carries a deletion that starts 42 bp upstream of the initiation codon in the 5'UTR and ends 191 bp before stop codon (Fig 6A), is predicted to be a protein null allele. Indeed, anti-Mst27D immunostaining of *Mst27D*<sup>cc</sup> testes did not reveal any specific signals (Fig 5D).

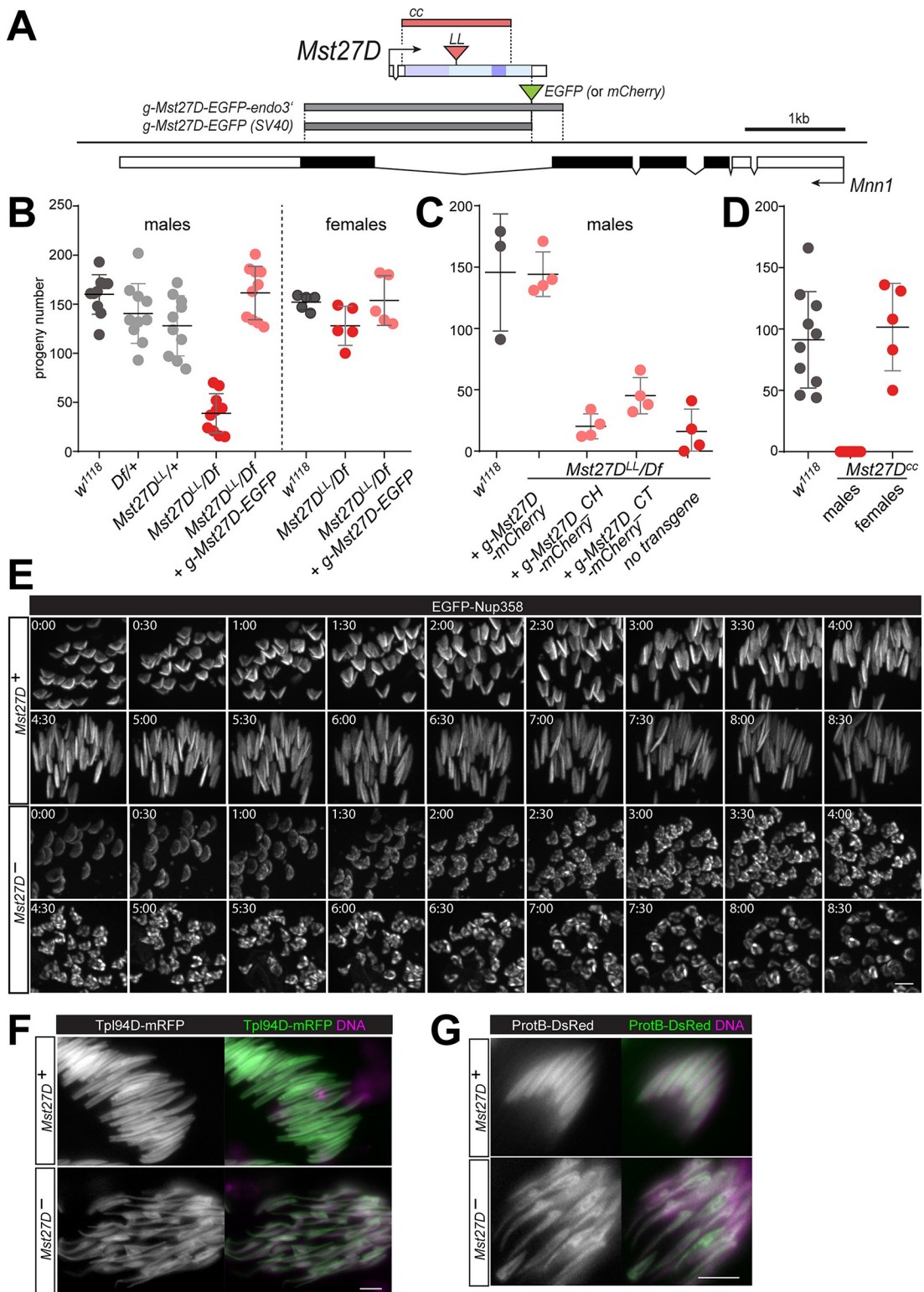

**Fig 6. Loss of *Mst27D* function is detrimental for male fertility and nuclear elongation in spermatids.** (**A**) Wild-type and mutant *Mst27D* alleles. *Mst27D* (top) is located within an intron of *Mnn1* (bottom). The *Mst27D^LL* allele carries a piggyBac insertion (red triangle) within the coding region, and the *Mst27D^cc* allele an intragenic deletion (red box), eliminating start codon and the majority of the coding region. Genomic regions present in the indicated transgenes are shown as well (grey boxes). (**B-D**) Fertility of *Mst27D* mutants. Males or females of the indicated genotypes were mated with *w^1118* flies and the number of resulting

adult F1 progeny flies was counted. *Df(2L)ade3* (*Df*) deletes *Mst27D*. The results of replicate crosses are indicated by filled circles; averages (+/- s.d.) are indicated as well. (**E**) Time-lapse imaging of nuclear elongation in *Mst27D^cc^/ Mst27D^LL^* mutants (*Mst27D^-^*) and in *Mst27D^cc^/ +* controls (*Mst27^+^*) expressing EGFP-Nup358. Still frames are displayed with nuclear clusters at the indicated time points (h:min). (**F,G**) Abnormal nuclear shape in *Mst27D* mutant spermatids during the stages characterized by nuclear Tpl94D (**F**) or ProtB (**G**). Whole mount preparations of testes expressing either *g-Tpl94D-mRFP* or *g-ProtB-DsRed* in a background that was either *Mst27D^cc^/ Df(2L)ade3* (*Mst27D^-^*) or *Mst27D^cc^/ +* (*Mst27^+^*) were labeled with a DNA stain. Single optical sections display spermatid nuclei at high magnification. Scale bars = 5 μm.

To assess whether *Mst27D* function is required for development to the adult stage, the progeny resulting from an *inter se* cross of flies with *Mst27D^cc^* over a balancer chromosome (*CyO*) was analyzed. *Mst27D^cc^* homozygotes comprised 34% of all eclosing flies, indicating that *Mst27D* function is not required for development to the adult stage. *Mst27D^LL^* homozygosity was also found to be compatible with development to the adult stage. Flies homozygous for *Mst27D^LL^* or *Mst27D^cc^* did not display obvious morphological abnormalities.

To analyze effects on fertility without potential compounding effects of second-site mutations potentially present on the *Mst27D^LL^* and the *Mst27D^cc^* mutant chromosomes, we crossed the alleles over the deficiency *Df(2L)ade3*, which deletes the *Mst27D* gene. F1 hemizygous *Mst27D* mutant males and females were crossed to *w^1118^* flies and the number of resulting adult F1 progeny was determined. Thereby, hemizygous *Mst27D^LL^* males were found to have a severely reduced fertility (Fig 6B and 6C). Average male fertility was about 20% of control (*w^1118^*) fertility. In contrast, female fertility was not reduced in *Mst27D^LL^* hemizygotes (Fig 6B). Comparable results were obtained with *Mst27D^cc^* (Fig 6D). As residual *Mst27D* mutant male fertility appeared to be age-dependent, the apparent difference between *Mst27D^LL^* and *Mst27D^cc^* males might reflect age variation in test males rather than differences in allele strength.

*Mst27D* is within an intron of *Menin 1* (*Mnn1*) (Fig 6A), which encodes a subunit of the *Drosophila* COMPASS-like complex [61,62]. Flies homozygous for an *Mnn1* protein null mutation were reported to be viable and fertile [63,64]. Thus, the severe reduction of male fertility that was observed in *Mst27^cc^* and *Mst27^LL^* mutants is unlikely due to effects of the alleles on *Mnn1* function. To exclude effects on *Mnn1*, we crossed *g-Mst27D-EGFP* into hemizygous *Mst27D^LL^* males. The transgene restored normal fertility to hemizygous *Mst27D^LL^* males (Fig 6B). Male fertility was also restored when *g-Mst27D-mCherry* was crossed into hemizygous *Mst27D^LL^* males, but not with the transgenes expressing only parts of Mst27D (*g-Mst27D_CH-mCherry* and *g-Mst27D_CT-mCherry*) (Fig 6C). Overall, our analyses demonstrate that *Mst27D* is crucial for male fertility.

To resolve whether loss of *Mst27D* compromises spermatogenesis, mutant testes were characterized microscopically using both fixed and live preparations. Until after meiosis, abnormalities were not detectable. Moreover, time-lapse imaging of transheterozygous *Mst27D^cc^/ Mst27D^LL^* mutants expressing *g-EGFP-Nup358* failed to reveal abnormalities during NE polarization and the subsequent polarization of the spermatid cysts, which clusters all the haploid nuclei at one end, with basal bodies docked in the center of the NPC-containing NE hemisphere and consistently oriented towards the other end of the cyst. However, during the following stages of nuclear elongation that are accompanied with Mst27D accumulation in wild-type spermiogenesis, *Mst27D* mutants displayed obvious defects (Fig 6E). In the mutants, nuclear elongation was limited and EGFP-Nup358 adopted a patchy distribution on the NE, while controls presented the normal spatial NPC-NE transformation from hemisphere to stripe along the elongation axis (Fig 6E).

The abnormality of nuclear elongation in *Mst27D* mutants appeared to vary from cyst to cyst. Nuclear elongation appeared to proceed to a variable extent, but the axis of elongation

was rarely straight and the nuclear diameter usually varied along the axis. For a statistical comparison of control and mutant cysts at comparable stages, we analyzed fixed testes expressing marker transgenes for spermatid staging. A first marker transgene was *g-Tpl94D-mRPF* [65], expressing red fluorescent transition protein Tpl94D under control of *Tpl94D cis*-regulatory sequences. A second marker transgene, *g-ProtB-DsRed*, expressed red fluorescent Protamine B (ProtB) under control of ProtB *cis*-regulatory sequences [66]. For histone replacement during normal spermiogenesis, Tpl94D and ProtB accumulate sequentially in the spermatid nuclei during the early and late canoe stages, respectively [2]. While Tpl94D accumulation is transient, ProtB remains until after fertilization. Average number of spermatid cysts positive for either Tpl94D-mRFP or ProtB-DsRed were similar in *Mst27D* mutants and control testes (mean number per testis tube with Tpl94D-mRFP signals = 19.1 +/- 3.6 s.d. in control and 19.7 +/- 3.2 in *Mst27D* mutants, n = 10 for each genotype; mean number with ProtB-DsRed signals = 23.5 +/- 4.6 s.d. in control and 26.5 +/- 4.8 in *Mst27D* mutants, n = 6 and 7 for control and *Mst27D* mutant, respectively). The spatial arrangement of cysts differing in nuclear marker protein accumulation and DNA signal compactness also supported the conclusion that the histone replacement process proceeds without gross temporal perturbations in *Mst27D* mutant testes. As in sibling control testis tubes, clusters of spermatid nuclei positive for either His2Av-mRFP, Tpl94D-mRFP or ProtB-DsRed were spatially ordered within the distal region with increasing compactness of the DNA signals. While histone replacement proceeded in spermatid nuclei of *Mst27D* mutants, nuclear shapes were highly abnormal. Instead of a normal canoe form, Tpl94D-mRFP positive nuclei in *Mst27D* mutants presented a far more variable, tadpole-like spatial appearance (Fig 6F). In controls, only 1.4% of the Tpl94D-mRFP positive cysts contained nuclei with irregular non-canoe shapes (n = 191 cysts from 10 testes). In contrast, 98% of the Tpl94D-mRFP positive cysts in *Mst27D* mutants displayed abnormal nuclear shapes (n = 197 cysts from 10 testes). Similar findings were made with the ProtB-DsRed marker (Fig 6G). In controls, only 1.4% of the ProtB-DsRed positive cysts contained nuclei with irregular non-canoe shape (n = 141 cysts from 6 testes), while 95% of the ProtB-DsRed positive cysts displayed abnormal nuclei in *Mst27D* mutants (n = 159 cysts from 7 testes). The low frequency of cysts with Tpl94D-mRFP or ProtB-DsRed and apparently normal nuclear shapes in *Mst27D* mutants (2 and 5%, respectively) appeared in line with their residual male fertility.

Analogous to male fertility, nuclear elongation was restored back to normal in *Mst27D* mutants expressing *g-Mst27D-mCherry*, while *g-Mst27D_CH-mCherry* and *g-Mst27D_CT-mCherry* did not result in normal nuclear elongation (S9B-S9D Fig). The Mst27D fragments expressed from the latter two transgenes in males with *Mst27D* function also displayed a subcellular localization (S9A Fig) that was either severely (in case of CH) or subtly (in case of CT) distinct from that of the full length protein (S6 Fig), similar as also evident after expression in S2R+ cells (Fig 3A and 3C).

In summary, our phenotypic characterizations demonstrated that *Mst27D* function is required for normal nuclear elongation in spermatids.

## Mst27D is required for formation of the MT bundle of the dense complex

The conspicuous strong bundle of MTs in the DC, initially revealed by ultrastructural analyses [20], was proposed to be crucial for nuclear elongation. Since our findings argued for Mst27D functioning as a linker between DC-MTs and the NPC-NE, a careful comparison of these DC-MTs in wild type and *Mst27D* mutants appeared of interest. While our immunofluorescent labeling with anti-tubulin antibodies after fixation with either formaldehyde or methanol resulted in weak signals in the DC, possibly because of accessibility problems, live imaging

with testes expressing either α- or β-tubulin tagged with GFP generated considerably stronger signals in control spermatids with *Mst27D*[+] function. MT organization revealed by live imaging of spermatids appeared to be somewhat distinct from that reported based on immunofluorescent analyses with fixed testes [67]. Moreover, live imaging revealed the temporal dynamics of MT organization in the DC. Intense cyst motility during this phase resulted in frequent displacement of the clustered nuclei out of focus and field of view, precluding successful tracking of individual spermatid nuclei throughout the complete nuclear elongation process. However, MT behavior in the DC of control spermatid cysts was revealed with multiple movies, covering 4–8 hours each. Nuclear elongation appeared to start at around 6 hours after the end of telophase II and lasted for approximately 10–14 hours (S10A Fig and S3 Movie).

EGFP-β-tubulin signals immediately after completion of M II (S10A Fig, t = 0:00) were most prominent on the centrosome and weak on an aster emanating from the centrosome and a coexisting cytoplasmic MT network, similar as observed in fixed cells [67]. Around 2–3 hours after M II completion, a single prominent straight MT bundle per spermatid appeared in the cytoplasm (S10A Fig, t = 2:00). This bundle, which has not been mentioned in the study with fixed cells [67], did not seem to be nucleated by the centrosome/basal body, which was docking on the NE during this period. Thereafter, extension of the tail MTs away from the basal body started (S10A Fig, t = 3:40). Concomitantly, periodic waves of contraction started to move across the cysts at intervals of about 20 minutes, while cyst polarization was observed. Spermatid nuclei were gradually redistributed to one end of the cyst and tail MTs extended towards the other end. Accompanied by the contraction waves, EGFP-β-tubulin signals became increasingly stronger in the region of the DC and nuclear elongation proceeded (S10A Fig, t = 6:39–19:18).

To characterize MT formation near nuclei during the elongation process in further detail, we performed time-lapse imaging with spermatid cysts expressing His2Av-mRFP in addition to α- or β-tubulin tagged with EGFP (Fig 7A–7C). Both EGFP-tubulin fusions yielded comparable results. At the start of nuclear elongation (around 6–10 hours after the end of M II), several MT bundles appeared extending away from the basal body and surrounding the spermatid nucleus similar to the ribs of an umbrella (Fig 7B and S4 Movie). Over time, the angle between the MT ribs decreased (as in a closing umbrella) and the MT bundles fused progressively into fewer and more prominent MT bundles, and finally into a single massive bundle (Fig 7B and 7C and S5 Movie). In parallel, an increasing deformation of the spherical nuclei occurred inside the MT ribs, resulting in nuclear extension eventually along the single prominent DC-MT bundle (Fig 7A–7C). A subtle but reproducible enrichment of the His2Av-mRFP signals in the subnuclear region next to the MT bundles accompanied nuclear elongation (Fig 7A). Time-lapse imaging of spermatid cysts expressing EGFP-α-tubulin and Mst27D-mCherry revealed extensive co-localization in the DC region. During the progressive fusion of MT bundles in the region of the DC that accompanied nuclear elongation, Mst27D-mCherry was clearly enriched on the MT bundles (Fig 7D and S6 Movie).

Our observations in control spermatids suggested that nuclear elongation is driven by the progressive bundling of MTs into increasingly robust and longer bundles that culminated in the formation of a single strong extended bundle in the DC. As the MT bundles of the DC extended away from the basal body, the bundles are extended likely by growth at distal MT plus ends. However, recent studies have reported the presence of γ-tubulin not only at the basal body but also at the opposite acrosomal end of the elongating spermatid nucleus, although at far lower levels [67–69]. Thus, sliding apart of hypothetical anti-parallel MTs in the DC by tetrameric Kinesin-5 motor proteins might be an alternative potential mechanism for nuclear elongation. By live imaging of spermatids expressing GFP-tagged tubulin, we did not detect outgrowth of MTs away from the acrosomal end. In an attempt to visualize MT

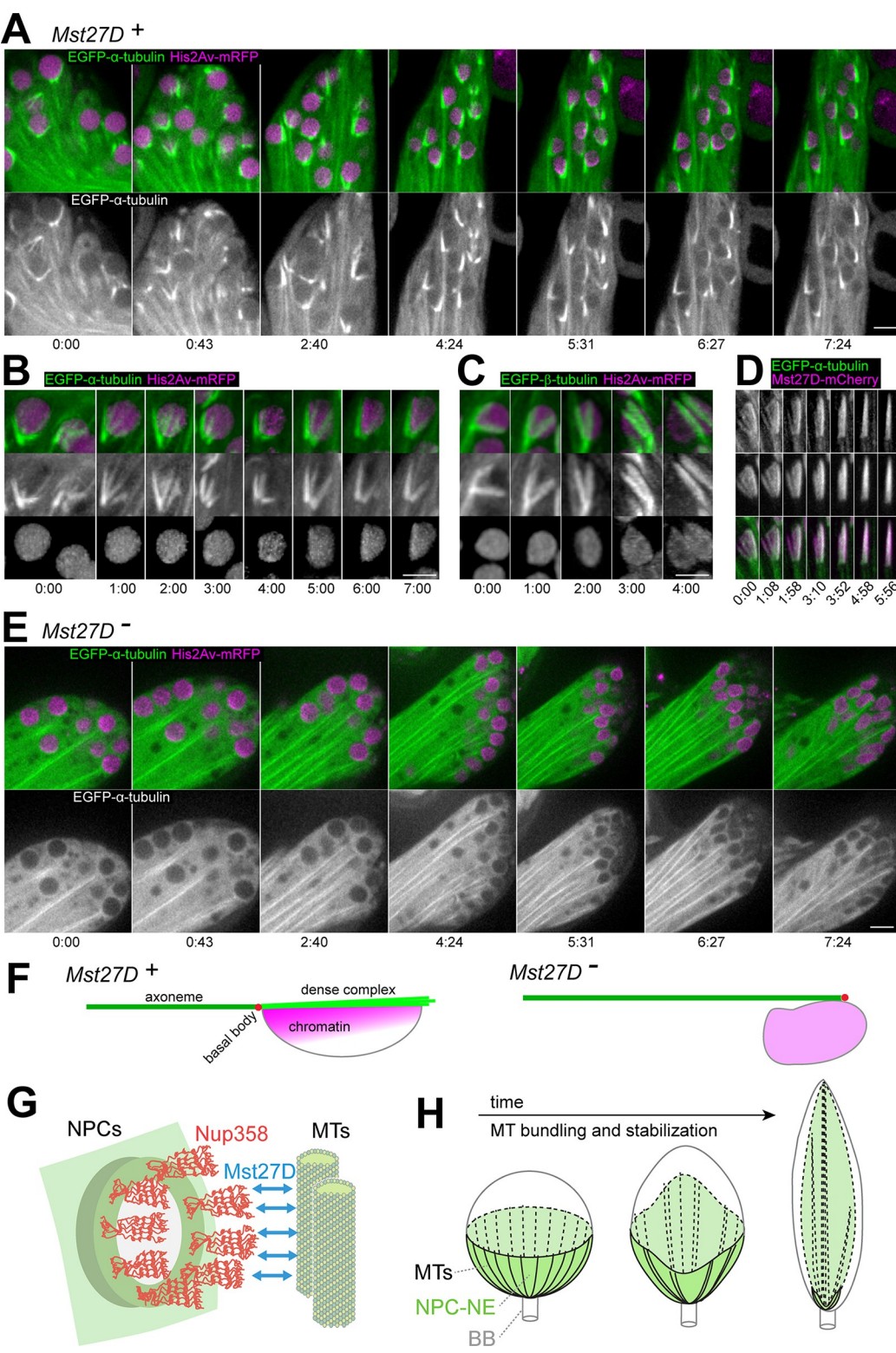

**Fig 7. *Mst27D*+ function is required for formation of the dense complex MT bundle.** (**A**-**E**) MT organization in the DC during nuclear elongation was analyzed by time-lapse imaging of spermatids with *Mst27D* function (*Mst27*+) (A-D) or without (*Mst27D*−) (E), expressing the indicated fluorescent proteins (His2Av-mRFP, EGFP-α-tubulin, EGFP-β-tubulin and Mst27D-mCherry). Still frames display regions with clustered spermatid nuclei (A and E) or individual nuclei at high magnification (B-D) with time indicated (h:min). Scale bars = 5 μm. (**F**) Cartoon summarizing MT and chromatin

organization accompanying nuclear elongation in control and *Mst27D* mutants. (**G**) Mst27D function during nuclear elongation. The N-terminal CH domain of Mst27D binds to MTs and its C-terminal domain binds to the C-terminal region of Nup358 that is estimated to be located within the cytoplasm about 60 nm away from the NE, as 8 x 5 copies of Nup358 form the NPC's cytoplasmic filaments. Mst27D forms dimers (or even higher oligomers) and thus might also function as an MT bundling protein (not illustrated). (**H**) Working model for formation of the DC. After docking of the basal body into the center of the NPC-NE hemisphere in early round spermatids, perinuclear MTs are nucleated, forming a radially symmetric array. Mst27D accumulation results in the binding of the MTs to NPCs. The MT bundling activity of Mst27D (or other MT associated proteins) breaks up the radial symmetry by progressive reorganization of the perinuclear MTs into a single bundle. Bundling is also proposed to promote stabilization and growth of the MTs within the bundles, presumably by increasing the local concentration of regulators of MT dynamics. MT bundling and extension in combination with attachment to the NPC-NE transform the shape of the nucleus from spherical to elongated needle shape.

growth in the DC, we analyzed spermatids expressing Eb1-tdGFP by time-lapse imaging. Moving comet-like structures highlighted by fluorescent versions of EB1 family proteins have successfully visualized growing MT plus ends in many studies. However, within the region of the DC, Eb1-tdGFP comets were not evident (S10C Fig). Instead, high, uniform and temporally stable Eb1-tdGFP signals were present throughout the entire DC, while far weaker moving comets were detectable in other regions as previously reported [70]. After photo-bleaching of Eb1-tdGFP in the central region of the DC, signals recovered within less than one minute uniformly throughout the bleached region (S10C Fig). Importantly, recovery was not accompanied by movement of Eb1-tdGFP comets across the bleached region. Thus, the high Eb1-tdGFP signals in the DC most likely reflect highly dynamic MT lattice binding. Analogous experiments for analysis of fluorescence recovery after photobleaching (FRAP) with spermatids expressing EGFP-α-tubulin, revealed far slower recovery in comparison to Eb1-tdGFP, indicating that DC-MTs are stable (S10C Fig).

To address the role of MT dynamics for DC elongation, demecolcine, an MT inhibitor, was added to spermatid cysts expressing EGFP-α-tubulin just before the start of time-lapse imaging. Demecolcine was added at a concentration that readily eliminated dynamic MTs like those assembling the meiotic division spindles, as also demonstrated previously [71]. However, the EGFP-α-tubulin signals in the DC were not weakened substantially by demecolcine, confirming that DC-MTs are stable, similar to those in the axoneme [67]. However, demecolcine inhibited the formation of a single extended MT bundle in the DC, as well as nuclear elongation (S10B Fig), suggesting that these processes depend on dynamic MT polymerization.

To assess the role of Mst27D for the organization of MT bundles in the DC, we analyzed *Mst27D* mutants (*Mst27D^cc^/ Df(2L)ade3*) expressing EGFP-α-tubulin and His2Av-mRFP. Interestingly, time-lapse imaging indicated that prominent MT bundles failed to form adjacent to the nucleus in *Mst27D* mutant spermatids during the stages of the abnormal nuclear elongation (Fig 7E and 7F and S7 Movie). Position and shape of the nuclei became increasingly abnormal in the *Mst27D* mutant spermatids (Fig 7E and 7F) compared to control (Fig 7A and 7F). In *Mst27D* mutant spermatids, the axoneme was observed to bypass the nucleus laterally, and the normal polarized enrichment of the His2Av-mRFP signal was not evident. In conclusion, the *Mst27D* mutant phenotype clearly revealed that *Mst27D⁺* function is required for the MT bundling process that normally proceeds in the DC. While still attempted, residual nuclear extension did not occur in a strictly linear direction in the mutants, indicating that the DC serves as a stiff rod that guides nuclear elongation during normal spermiogenesis.

The spermiogenic processes following nuclear elongation, i.e., NPC-NE shedding and spermatid individualization, were also abnormal in *Mst27D* mutants (S11 Fig), presumably at least in part as a secondary consequence of the defective nuclear elongation. Formation of investment cones (ICs), which mediate sperm individualization, was clearly observed in *Mst27D* mutants by staining of F-actin with fluorescent phalloidin (S11 Fig). Moreover, before

completion of IC assembly, NPC-NE shedding was observed in hemizygous $Mst27D^{LL}$ mutants expressing Nup58-EGFP (S11 Fig). However, while in controls Nup58-EGFP was stripped off from the spermatid nuclei completely (except for a remaining very weak basal dot), substantial amounts of Nup58-EGFP remained on the NE around the spermatid nuclei of *Mst27D* mutants (S11 Fig), indicating partial individualization defects.

## Mst27D and the sperm-specific SUN domain protein Spag4 function independently

Our results indicated that Mst27D, as a linker between NPCs and MTs, is required for proper nuclear elongation along the straight line defined by the DC MT bundle. In mammalian spermiogenesis, nuclear shaping is thought to depend on connections between NE and MTs that are provided by LINC complexes containing testis-specific SUN domain proteins [26,72]. *Drosophila* Spag4, a SUN domain protein expressed exclusively during spermatogenesis, was reported to display a localization similar to that of Mst27D. In spermatocytes, Spag4 localization is still symmetric throughout the NE but after NE polarization it is confined to the hemispherical NPC-NE, where it is required for dynein-dynactin recruitment to this NE region in early round spermatids. During nuclear elongation, Spag4 remains enriched on the NPC-NE although with decreasing intensity [73]. To evaluate potential hierarchies in the localization of Spag4 and Mst27D, we analyzed Mst27D-EGFP localization in *spag4* mutants, as well as Spag4-EGFP localization in *Mst27D* mutants (S12 Fig).

In *spag4* mutants, Mst27D-EGFP displayed a normal localization during NE polarization in early round spermatids and also during the subsequent nuclear elongation, which appeared to be normal (S12A Fig). Nuclear morphology was still normal even in late *spag4* mutant spermatids soon after NPC-NE and Mst27D-EGFP shedding at the start of individualization. However, in very late spermatids, the needle shaped nuclei displayed a highly abnormal curled morphology in *spag4* mutants (S12A Fig), as reported [73].

The localization of Spag4-EGFP was not affected in *Mst27D* mutants (S12B Fig). In both control and *Mst27D* mutants, Spag4-EGFP localization was slightly distinct from that of Mst27D-EGFP in early round spermatids. While both Mst27D-EGFP and Spag4-EGFP were restricted to the NPC-NE (S12B Fig), Spag4-EGFP was most strongly enriched in the basal body region, which was largely devoid of Mst27D-EGFP (S12B Fig). This preferential localization of Spag4-EGFP on the basal body increased further during nuclear elongation in controls and also during the corresponding stages in *Mst27D* mutants (S12B Fig).

In conclusion, the proteins Mst27D and Spag4, which both connect NE with MT structures in spermatids, localize independently and provide distinct functions. While Spag4 maintains the connection between needle shaped nucleus and basal body/axoneme during the final stages of spermiogenesis [73], Mst27D mediates the attachment of the NPC-NE to the DC-MTs to permit normal nuclear elongation.

## Discussion

During spermiogenesis, the cell nucleus is extensively remodeled in many animal species. In *D. melanogaster*, nuclear volume is reduced 200-fold and nuclear shape is transformed from spherical to elongated needle during spermatid differentiation. Remarkably, nuclear remodeling is accompanied by striking changes in the distribution of NPCs within the NE of spermatid nuclei, not just in *D. melanogaster* [11,20], but also in humans and other mammals [21–23]. During nuclear elongation, the NPC-NE, i.e., the NE region that still contains NPCs, is intimately associated with a strong MT bundle that is assembled in the so-called dense complex (DC) of *D. melanogaster* spermatids [11,20]. Our present work demonstrates that the

spermatid-specific Mst27D protein links NPCs to the MT bundle of the DC. The N-terminal CH domain of Mst27D binds to MTs and its C-terminal part (CT) interacts directly with the C-terminal region of Nup358 (Fig 7G). Moreover, our phenotypic characterization of *Mst27D* mutants demonstrates that the progressive bundling and extension of MTs in the DC is dependent on Mst27D and that this process drives the normal straight elongation of spermatid nuclei (Fig 7H).

The NPC redistribution in *D. melanogaster* spermatids includes two symmetry breaking episodes. The first is observed in early round spermatids, when the initial largely symmetric localization of NPCs all around the spherical nucleus is converted into a hemispherical NPC distribution during the process designated here as NE polarization. The second symmetry break occurs around the start of nuclear elongation, when the hemispherical NPC distribution is further restricted into essentially a line along the axis of nuclear extension. Our time-lapse imaging of Nup58-EGFP and EGFP-Nup358 has delivered an improved temporal resolution of these processes, which were previously studied only with fixed samples primarily by EM [11,20,67]. The conversion from spherical to hemispherical NPC distribution was observed to start about one hour after completion of the second meiotic division and was accomplished within about one hour. The change from hemispherical to linear arrangement was initiated about six hours after M II completion and required approximately six hours. Eventually, after completion of nuclear elongation in *D. melanogaster* spermatids, NPCs are eliminated from the NE. We demonstrate that this NPC-NE shedding process is completed within about an hour just after the onset of nuclear Protamine B accumulation and of IC assembly before spermatid individualization.

We emphasize that the first symmetry breaking, NE polarization, is completed before the first nuclear elongation stage of the classical scheme [12] shown in Fig 1B. The initial NE polarization results in the presence of NPCs in a hemisphere that docks the centrosome/basal body in its center [74]. Only thereafter, with the second symmetry breaking, a reorientation of the NPC distribution proceeds, as illustrated in Fig 7G, which places the acrosomal region and the basal body at opposite ends of the NPC-NE region (similar as in the first drawing in Fig 1B).

A dependence of the first symmetry breaking (NE polarization) on Mst27D initially seemed conceivable based on the pattern of expression of the *g-Mst27D-EGFP* transgene [32]. However, anti-Mst27D immunostaining, as well as our analysis of *g-Mst27D-EGFP*-endo3' and *Mst27D* null mutant testes, disproved this notion. Endogenous Mst27D accumulation appears to occur after NE polarization, which proceeded normally in the *Mst27D* null mutants. Thus, the molecular basis of this intriguing NE polarization process remains to be clarified. Similar NE polarization has recently been shown to proceed in human and bovine zygotes during the pronuclear stage, by a mechanism relying on NPCs recruiting dynein-dynactin in combination with dynein's motor activity towards the minus ends of MTs in asters nucleated by the centrosome in between the pronuclei [75]. Moreover, physical connections between NPCs and MTs via dynein-dynactin have also been observed in cultured mammalian cells at the start of mitosis [76,77]. Interestingly, dynein-dynactin is strongly enriched on the NE and confined to the hemispherical NPC-NE after NE polarization in early *D. melanogaster* spermatids [78]. This NE-associated dynein-dynactin pool is thought to promote the docking of the basal body at the central pole of hemispherical NPC-NE by pulling on MTs nucleated by the centriole/basal body [29,73,74]. However, rather than via NPCs, recruitment of dynein-dynactin to the NPC-NE is mediated by Spag4, a male germline-specific SUN domain protein [73]. In *spag4* mutants, dynein-dynactin is no longer enriched on the NE of early spermatids and basal body docking within the NE is defective [73]. Therefore, our observation that NE polarization, as reported by Mst27D-EGFP, is normal in *spag4* mutants indicates that this process is not driven

by dynein-dynactin motor activity. Moreover, NE polarization was also not affected after addition of high concentrations of the MT drug demecolcine before time-lapse imaging. Given this evidence against NE polarization by NPC-associated dynein-dynactin interacting with centrosomal MTs, alternative mechanisms deserve consideration. Ultrastructural analyses might provide hints [11,20]. NE polarization in *D. melanogaster* spermatids is accompanied by a drastic reduction in the spatial separation between ONM and INM in the NPC-free region, as also in mammalian spermatids, where NPCs disappear from the anterior hemisphere of the nucleus concomitant with progressive posterior spreading of the acrosomal cap over the anterior NE [21–23,79]. Thus, NE polarization might involve unknown machinery that remodels the NE, including ONM-INM organization.

Mst27D is clearly crucial for the second symmetry breaking, the conversion of the hemispherical NPC-NE into a linear arrangement, defining the axis of nuclear elongation. Endogenous Mst27D accumulates after completion of the process that docks the centriole/basal body at the center of the hemispherical NPC-NE. After this docking, pericentriolar material associates around the proximal basal body, forming the so-called centriolar adjunct, which has enhanced MT nucleation activity [67,74]. We propose that Mst27D establishes physical linkages between the NPC-NE and the perinuclear MTs of the aster emanating from the centriolar adjunct (Fig 7G and 7H). The spatial separation between NPCs and the closest MTs of the dense complex that is estimated to be around 60 nm by EM [11,20,67] is entirely compatible with physical linkage by Mst27D because this linker protein binds to the C-terminal region of Nup358, which is thought to extend up to 60 nm into the cytoplasm [37]. Nup358 is clearly required for normal enrichment of Mst27D on the NE, as indicated with our experiments involving Nup358 elimination by deGradFP.

Beyond physical linkage between NPCs and MTs, bundling of MTs is of major importance for nuclear elongation according to our proposed model (Fig 7G and 7H). Progressive bundling of the perinuclear MTs, as clearly revealed by our time-lapse imaging, is proposed to break the initial radial symmetry of the MTs emanating from the centriolar adjunct. Progressive MT bundling transforms the radial array into fewer and eventually into a single longitudinal MT bundle that marks a meridional line from basal body to acrosome. The MT bundling proceeded over a period of about 10–12 hours. We propose that bundling results in stabilization of the MTs within the bundles, explaining the longitudinal extension of the dense complex. Moreover, the persisting dynamic association between NPC-NE and MT bundles that is mediated by Mst27D during the bundling process is proposed to promote the elongation of the spermatid nucleus (Fig 7G and 7H).

We propose that MTs in the final single bundle within the DC have a uniform polarity, since they are nucleated by the centriolar adjunct. The appearance of the MT bundles during progressive bundling in the DC is consistent with this notion. MT bundles emanate with a blunt end from the caudal side of the nucleus and taper off towards the acrosomal side. Thus, nuclear elongation is likely promoted by the progressive elongation of the MTs within the DC. While we consider our evidence obtained from time-lapse imaging with GFP-tagged tubulin to be highly intriguing, we emphasize that alternative proposals for MT formation and organization in the DC have been proposed based on earlier analyses with fixed testes [20,67]. A future definitive clarification of MT polarity within the DC is clearly important for a detailed understanding of nuclear elongation. Our attempts to study MT outgrowth during DC formation by time-lapse imaging of EB1-tdGFP failed to reveal moving comets (i.e., growing MT plus ends) above a high uniform signal all along the MT bundles.

Mst27D might not just function as an NPC-MT linker but also as an MT-MT cross-linker that thereby promotes MT bundling. Each NPC is estimated to contain 40 copies of Nup358 (eight pentamers) [37]. In principle, therefore, multiple MTs can be recruited via Mst27D to

each NPC. Moreover, the CT region of Mst27D promotes the formation of dimers (or perhaps of even larger oligomers) according to our co-immunoprecipitation experiments. Thus, the two CH domains of an Mst27D dimer might be able to bind to two distinct MTs, resulting in cross-links and in MT bundling at sufficiently high concentrations. Extensive MT bundling was clearly apparent in S2R+ cells expressing high levels of Mst27D-EGFP or Mst27D-mCherry. MT bundling was also obtained with the chimeric protein Mst27D_CH-Eb1_CT-EGFP, indicating that the MT bundling activity does not require the concentration of Mst27D by recruitment to NPCs. It remains to be established conclusively that the concentration of endogenous Mst27D protein is sufficiently high for it to function as an MT bundler during nuclear elongation in spermatids. As a caveat, mCherry appears to have some residual potential for dimer formation, although far lower than EGFP [48]. Weak dimerization via mCherry (or even stronger via EGFP) might enhance the bundling activity of Mst27D, in particular when expressed at very high concentrations.

Although Mst27D promotes potentially all of the crucial interactions required for dense complex formation and nuclear elongation, it seems likely that other factors contribute. According to EM analyses [11,20], the cytoplasmic region between NPC-NE and an additional double membrane system that is present around the nucleus in early round spermatids has a distinct dense texture already before perinuclear MTs fill this space. This "perinuclear plasm" [11] presumably contains material other than Mst27D, because the level of this protein is still very low during these initial spermatid stages. Thus, additional proteins are likely enriched in this region, perhaps regulating MT dynamics, bundling and stabilization. The yet uncharacterized EB1 family protein encoded by *CG15306* is an interesting candidate with potentially redundant function. *Mst27D* appears to have arisen by retroposition from *CG15306* [80], a gene recently also implicated in the evolution of the *Dox* meiotic drive system in *D. simulans* [81]. *Dm CG15306* displays testis-specific expression like *Mst27D*, and the additional uncharacterized EB1 family members encoded by *CG32371* and *CG2955*. Future analysis of the evolutionary history of *Mst27D* and the other EB1 family genes (Fig 1C), as well as the extent of functional redundancies among these genes, is likely of considerable interest.

A contribution to nuclear elongation by players other than Mst27D is also suggested by the residual elongation that can be observed in *Mst27D* null mutants. The periodic contraction waves that run down the cysts during the stages of cyst polarization and nuclear elongation, as well as the individualization process are likely contributors to residual nuclear elongation in *Mst27D* null mutants. Residual nuclear elongation varied considerably from cyst to cyst in *Mst27D* null mutants and was never as straight as in controls. Thus, the single strong MT bundle that is eventually present in the dense complex during normal spermiogenesis although not in *Mst27D* null mutants appears to provide robustness and directionality to nuclear elongation.

The dense complex with its MT bundle that characterizes nuclear elongation in *D. melanogaster* has often been suggested to correspond to the manchette that is assembled around elongating nuclei in mammalian spermatids. A main component of the manchette is a cylindric array of MT bundles that emanate from a perinuclear ring structure immediately below the acrosome cap and extend caudally beyond the partially elongated spermatid nuclei [30]. A contraction of the manchette diameter constrains the posterior part of the nucleus during nuclear elongation [30]. Interestingly, manchette formation is accompanied by the rapid disappearance of NPCs from the lateral NE, in particular in regions adjacent to the MT bundles [23]. Concomitantly, folds of the NE that have a very high density of NPCs are extended into the cytoplasm on the posterior side of the nucleus [23]. Formation of these NPC-NE folds (the so-called redundant NE) might thus be related to NPC-NE shedding in *D. melanogaster* spermiogenesis, except that the redundant NE is not stripped away from the nucleus completely, at

least in some mammalian species. Redundant NE formation is thought to result from manchette-mediated transport of NPCs towards the posterior, although this has not been observed directly. As both NPC-NE shedding in *D. melanogaster* and redundant NE formation in mammals are not understood at the molecular level, the extent of mechanistic similarity is presently far from clear. However, some notable differences between dense complex and manchette are apparent. The multiple MT bundles of the manchette at regular intervals all around the circumference of the elongating nuclei remain separate. In contrast, a single MT bundle is generated eventually by progressive bundling in the dense complex. The MT bundles of the manchette, which were proposed to be attached to the NE by LINC complexes [25,26], are next to NE that is free of NPCs [23]. In contrast, the single MT bundle in the dense complex is juxtaposed to the NPC-NE. The known *D. melanogaster* SUN and KASH domain proteins are not required for linkage of dense complex MTs to the NE [73]. Instead, Mst27D appears to establish direct linkage. A mammalian Mst27D homolog does not appear to exist.

Overall, our results demonstrate that Mst27D is of pivotal importance for the proper nuclear elongation in spermatids of *D. melanogaster*. Based on ultrastructural analyses, nuclear elongation was suggested 50 years ago to be promoted by the dense complex, a structure containing an elongated massive bundle of MTs closely associated with the NPC-NE. However, direct experimental evidence demonstrating the role of the dense complex for nuclear elongation has been missing so far. Moreover, although physical linkage between NPCs and MTs of the dense complex was suspected, the molecular basis for such an interaction was unknown and difficult to explain given the considerable spatial separation between NE and the closest MTs of the dense complex. Based on our functional characterization of Mst27D function, we propose a molecular explanation for the formation and function of the dense complex in nuclear elongation.

## Materials and methods

### *Drosophila* lines

The source of the mutant alleles and transgenes that were used for this study is given in S2 Table. Transgenes under control of *cis*-regulatory regions derived from the endogenous genomic loci are named here with the prefix "*g-*"(as in *g-EGFP-Nup358* for example). Standard crossing and meiotic recombination were used to generate strains with combinations of these mutant alleles and/or transgenes. The genotypes of the flies analyzed are described in detail in S3 Table, and additional explanations are provided in the following. Flies analyzed were usually raised at 25°C. $w^1$ or $w^{1118}$ were used as wild-type control unless stated otherwise.

Lines with *g-Nup58-EGFP* generated with a pCaSpeR4 construct have been described previously [82]. For generation of lines with the transgenes *g-Mst27D-mCherry*, *g-Mst27D-Dendra2*, *g-Mst27D_CH-mCherry*, *g-Mst27D_CT-mCherry* or *g-Mst27D-EGFP_endo3'*, the pCaSpeR4 constructs described further below were injected into $w^{1118}$ embryos (BestGene Inc., Chino Hills, CA, USA). The transgenic *g-EGFP-Nup358* line was obtained after microinjection of pattB_gEGFP-Nup358 (see below for plasmid description) into $y^1 w^{67} c^{23}$; *P{CaryP}attP40* embryos. A homozygous viable and fertile *g-EGFP-Nup358* stock was established.

The $Nup358^{12A002}$ mutant allele [83] has an A to T mutation at nucleotide position 1669 resulting in a premature stop codon (K557stop). The mutant Nup358 proteins that might be expressed from this allele do not include the C-terminal region required for Mst27D binding. $Nup358^{0175\text{-}G4}$ [84] contains a *piggyBac* insertion in the 5' UTR 12 bp upstream of the start codon. Trans-heterozygous $Nup358^{12A002}/ Nup358^{0175\text{-}G4}$ flies did not develop to the adult stage. However, with the *g-EGFP-Nup358* transgene present in this trans-heterozygous background, the mutants developed into viable and fertile adults.

For comparison of the localization of Nup358 and Mst27D, the line *w\*; P{w+, g-EGFP-Nup358} attP40/ CyO; PBac{w+, IT.GAL4}Nup358^0175-G4, P{w+, g-Mst27D-mCherry} III.7/ TM6B, Tb, Hu* was generated. Males without balancer chromosomes were used for whole mount preparations and live imaging (Fig 4A). Alternatively (Fig 5A), we used males without balancers arising in the stock *w\*; P{w+, g-Mst27D-mCherry} II.3/ CyO; emGFP-Nup358, PBac {y[+mDint2] = vas-Cas9}VK00027/ TM6B, Tb, Hu*.

To assess effects on the subcellular localization of Mst27D-mCherry after EGFP-Nup358 elimination by deGradFP (Figs 4B–4D and S5B), *g-Mst27D-mCherry* was first recombined with the *Nup358^0175-G4* allele and then combined with *g-EGFP-Nup358*. The genotype of the resulting fly stock was *w\*; P{w+, g-EGFP-Nup358} attP40/ CyO, Dfd-YFP; PBac{w+, IT.GAL4} Nup358^0175-G4, P{w+, g-Mst27D-mCherry} III.7/ TM6B, Tb, Hu*. In addition, the *Nup358^12A002* allele was combined with *exumP-Nslmb-vhh4-GFP*. The genotype of the resulting fly stock was *w\*; P{w+, exumP-NSlmb-vhhGFP4}attP40/ CyO; P{w+, FMRFa-EGFP.Tv}3, P{w+, UAS-myr-mRFP}2, Nup358^12A002/ TM6B, Tb, Hu*. The two stocks were then crossed to generate the analyzed trans-heterozygous *Nup358* mutant genotype *w\*; P{w+, g-EGFP-Nup358} attP40/ P{w+, exumP-NSlmb-vhhGFP4}attP40; PBac{w+, IT.GAL4} Nup358^0175-G4, P{w+, g-Mst27D-mCherry} III.7/ P{w+, FMRFa-EGFP.Tv}3, P{w+, UAS-myr-mRFP}2, Nup358^12A002* ("+deGrad" genotype). In addition by crossing the first stock with a third stock *w1118; P{w+, FMRFa-EGFP.Tv}3, P{w+, UAS-myr-mRFP}2, Nup358^12A002/ TM6B, P{w+, Dfd-EYFP}3, Sb Tb ca*, the additional analyzed control genotype *w\*; P{w+, g-EGFP-Nup358} attP40/ +; PBac{w+, IT.GAL4} Nup358^0175-G4, P{w+, g-Mst27D-mCherry} III.7/ P{w+, FMRFa-EGFP.Tv}3, P{w+, UAS-myr-mRFP}2, Nup358^12A002* ("-deGrad" genotype) was generated.

*Mst27D* mutant alleles were generated with CRISPR/Cas9. The plasmids pCFD5-sgRNA-Mst27D-1_4 and pCFD5-sgRNA-Mst27D-2_3 (see below) were injected individually into embryos of the stock *yw;;nos-cas9(III-attP2)* (BestGene Inc., Chino Hills, CA, USA). Resulting adult G0 males were crossed out singly with three *w\*; Gla/CyO* virgins. The primer pair LP043/LP044 (see S4 Table for oligonucleotide sequences), which annealed 86 bp upstream and 147 bp downstream of the two gRNA-targeting sites, respectively, were used for identification of deletions by PCR in genomic DNA isolated from a pool of some of the F1 progeny. Eventually, the *Mst27D^cc1-4* allele was established as a balanced stock. Molecular characterization by PCR and sequencing indicated the presence of an intragenic deletion. Sequences from 42 bp upstream of initiation codon until 191 bp before the stop codon were replaced by a novel short 7 bp sequence (5'-<u>TTAGAAT</u>-3'), resulting in a sequence across the breakpoints of 5'- . . . TACTCCTTACC<u>TTAGAAT</u>AAACGGCAAG- . . . 3'.

The dynamics of NPC-NE shedding in late spermatids (S7 Fig) was analyzed by time-lapse imaging with the genotype *w\*; P{w+, g-Nup58-EGFP}35B (12.4); P{w8, ProtB-DsRed-M1}50A (III)*. The transgene *g-ProtB-DsRed* [66] carries a genomic 4.2 kb fragment containing the Protamine B gene region and 1.3 kb of sequence upstream of the transcription start site and 1.2 kb of sequence downstream of the polyadenylation site.

For analyses with *g-Mst27D-EGFP_endo3'* in a *spag4* null mutant background, the alleles *spag4^1* and *spag4^6* were used in trans. These *spag4* alleles were generated by replacing the wild-type *spag4* gene with *w+* using ends-out homologous recombination [73]. The *spag4-GFP* transgene [73] was analyzed in the hemizygous Mst27D mutant background *Mst27D^cc1-4/ Df(2L)ade3*. The genotype *w\*; Mst27D^cc1-4/ +; P{w+, spag4-GFP} 3/ +* was used as wild type control (S12 Fig).

## Plasmids

The gRNA expression plasmids pCFD5-sgRNA-Mst27D-1_4 and pCFD5-sgRNA-Mst27D-2_3 were made using the vector plasmid pCFD5 [85] and the synthetic DNA fragments LP049

and LP050 (gBlocks Gene Fragments obtained from IDT, Leuven, Belgium) (for sequences see S4 Table). The synthetic DNA fragments and pCFD5 were digested with BbsI before ligation. An online tool (http://targetfinder.flycrispr.neuro.brown.edu/) was used for identification of gRNA targets sites. Each pCFD5 derivative contained two distinct gRNA sequences targeting different regions of *Mst27D*.

The plasmids pCaSpeR-gMst27D-mCherry, pCaSpeR-gMst27D_CH-mCherry, pCaSpeR-gMst27D_CT-mCherry, pCaSpeR-gMst27D-Dendra2 and pCaSpeR-gMst27D-EGFP-endo3' were made as follows. For pCaSpeR-gMst27D-mCherry, a fragment containing the *Mst27D* coding region and 1045 bp upstream sequences was amplified from $w^1$ genomic DNA with the primer pair LP017/LP018. The fragment was digested with BssHII and BglII and ligated into pUASt-mcs-mCherry [86] that had been digested with MluI and BglII. Sequencing of the resulting construct (pCaSpeR-gMst27D-mCherry) revealed the presence of a mutation that induces an aa sequence change (N92T) compared to the sequence predicted by the FlyBase reference genome. Sequencing of an independent PCR product amplified from $w^1$ genomic DNA with the primer pair LP011/LP021 revealed an identical single base change, indicating that it reflects a single nucleotide polymorphism present in the $w^1$ genome. Our cloning strategy eliminated the sequences containing the multimerized $UAS_{GAL4}$ binding sites and the minimal heat shock promoter that were present in the pUASt-mcs-mCherry vector. However, the 3' sequences downstream of the mCherry coding sequence were the SV40 terminator sequences derived from pUASt-mcs-mCherry. The final construct pCaSpeR-gMst27D-mCherry was therefore analogous to the g-Mst27D-EGFP transgene construct of Gärtner et al. (2019) [32] except that the mCherry rather than the EGFP coding sequence were fused C-terminally.

The construct pCaSpeR-gMst27D-mCherry was modified the generate pCaSpeR-gMst27D_CH-mCherry and pCaSpeR-gMst27D_CT-mCherry. For cloning of these derivatives, the synthetic insert DNA fragments CL342 and CL343, respectively, and the vector pCaSpeR-gMst27D-mCherry were digested with SpeI and BglII before ligation. A similar strategy was used for the generation of pCaSpeR-gMst27D-Dendra2. The synthetic DNA fragment CL344 and the vector pCaSpeR-gMst27D-mCherry were ligated after digestion with KpnI and XbaI.

To generate pCaSpeR-gMst27D-EGFP-endo3', pUASt-mcs-EGFP and pCaspeR-gMst27D-mCherry were both digested with XhoI and XbaI to produce an insert fragment from pCaspeR-gMst27D-mCherry and vector fragment from pUASt-mcs-EGFP that were ligated, yielding the cloning intermediate pCaSpeR-gMst27D-EGFP. The SV40 terminator sequences in this intermediate were replaced with genomic 3' sequences derived from the *Mst27D* locus that were obtained by amplification from $w^1$ genomic DNA with the primers CL396/CL397. The resulting DNA fragment comprising 288 bp of *Mst27D* 3' sequences downstream of the stop codon and pCaSpeR-gMst27D-EGFP) were digested with XbaI and BamHI before ligation which generated pCaSpeR-gMst27D-EGFP-endo3'.

For the construction of pattB_gEGFP-Nup358, the primer pair CL405/CL406 was used to amplify a fragment containing 827 bp of upstream sequences with the Nup358 5'UTR region, intron 1 and the beginning of exon 2 from $w^1$ genomic DNA. The PCR fragment was ligated as insert into pattB after digestion of insert and vector with BamHI and EcoRI, yielding the cloning intermediate 1 (pattB-Nup358_upstream). The EGFP coding sequence (736 bp) was amplified from pUASt-mcs-EGFP using the primer pair RAS42/CL385. The PCR product was then ligated into pattB-Nup358_upstream after digestion of insert and vector with EcoRI and NotI, yielding intermediate 2 (pattB-Nup358_upstream-EGFP). The plasmid pUbiqP_mCherry-Nup358-ry_2 (see below) was digested with NotI and SalI to generate an insert fragment that was ligated into pattB-Nup358_upstream-EGFP, which had been digested with NotI and XhoI, yielding the final construct pattB_gEGFP-Nup358.

The plasmids pMT-Mst27D-EGFP_bla, pMT-Mst27D_CH-EGFP_bla, pMT-Mst27D_CT-EGFP_bla, pMT-Mst27D-mCherry-bla and pMT-Mst27D_CH-EB1_CT-hl-EGFP_bla used for transfection of S2R+ cells were made as derivatives of pMT-bla [35]. In case of pMT-Mst27D-EGFP_bla, the primer pair CL349/CL231 was used for amplification of an Mst27D-EGFP fragment (1995 bp) from genomic DNA of g-Mst27D-EGFP flies [32]. The PCR fragment and pMT-bla were digested with BglII and NotI before ligation, yielding pMT-Mst27D-EGFP_bla. For construction of pMT-Mst27D-CH-EGFP_bla, an insert was produced by overlap PCR. The region including the sequences coding for the CH domain was amplified from pMT-Mst27D-EGFP_bla with the primer pair CL349/LP037 and the EGFP coding sequence was amplified from pMT-Mst27D-EGFP_bla with the primer pair LP038/CL231. A CH-EGFP fusion fragment was then produced using a mixture of these two fragments as template and the primer pair CL349/CL231 in the overlap PCR. The overlap PCR product was ligated into pMT-bla after digestion of vector and insert with BglII and NotI. For the generation of pMT-Mst27D-CT-EGFP_bla, the insert fragment was amplified from pMT-Mst27D-EGFP_bla with the primer pair LP039/CL231. The PCR product was then ligated into pMT-Mst27D-EGFP_bla after digestion of insert and vector with SpeI and NotI. For pMT-Mst27D-mCherry-bla, the plasmid pMT27D-CT-mCherry (vector) and pCaSpeR-Mst27D-EGFP-endo3' (insert) were digested with SpeI and NotI before ligation. For pMT-Mst27D_CH-EB1_CT-hl-EGFP_bla, the synthetic DNA fragment CL381 was ligated into pMT-hl-EGFP-bla after digestion with BglII and NotI.

For the construction of pUbiqP_mCherry-Nup358-ry_2, the primers CL350 and CL351 were annealed to produce a linker that was inserted into pUbiqPry [87] that had been digested previously with Asp718 and SpeI. Thereby, the first cloning intermediate (pUbiqPry-linker_2) was obtained. The mCherry coding sequence was amplified from pUASt-mcs-mCherry with the primer pair CL352/CL353. After digestion of the resulting fragment and pUbiqPry-linker_2 with SpeI and NotI, a ligation generated the second cloning intermediate (pUbiqP-mCherry-ry_2). Thereafter, pUbiqP-mCherry-ry_2 and the Nup358 expressed sequence tag (EST) plasmid pOT2_LD43045 (Drosophila Genome Resource Center, Indiana University, Bloomington, Indiana, USA) were digested with HpaI and SacI for subsequent ligation, yielding cloning intermediate 3 (pUbiq-mCherry-Nup358-4-ry_2). In a next step, pUbiq-mCherry-Nup358-4-ry_2 and the additional Nup358 EST plasmid pOT2_LD24888 (Drosophila Genome Resource Center, Indiana University, Bloomington, Indiana, USA) were digested with HpaI and EcoRI, and ligated to generate cloning intermediate 4 (pUbiq-mCherry-Nup358-2/3/4-ry_2). Finally, an additional Nup358 fragment was amplified with CL358 and CL359 from pOT2_LD24888 and ligated into pUbiq-mCherry-Nup358-2/3/4-ry_2 after digestion of fragment and vector with NotI and EcoRI, resulting in the final construct pUbiq_mCherry-Nup358-ry_2.

To generate plasmids for expression of a series of C- and N-terminal Nup358 truncations, a mutagenic amplification method [88] was used to engineer NheI target sites into pUbiqP_mCherry-Nup358-ry_2 at suitable positions within the Nup358 coding region. The plasmid pUbiqP_mCherry-Nup358-Nhe1 was generated with this method using the primer LP023 and had the introduced NheI site around the codon of aa residue 2342 of Nup358, pUbiqP_mCherry-Nup358-Nhe2 generated with the primer LP024 around aa residue 1970, pUbiqP_mCherry-Nup358-Nhe3 generated with the primer LP025 around aa residue 1498, pUbiqP_mCherry-Nup358-Nhe4 generated with the primer LP026 around aa residue 1269, and pUbiqP_mCherry-Nup358-Nhe5 generated with the primer LP027 around aa residue 830. To create plasmid derivatives for expression of N-terminal fragments, the five pUbiqP_mCherry-Nup358 versions with introduced NheI sites were digested with NheI and AvrII. Thereafter, a linker obtained by annealing the primers LP030 and LP031 was ligated into the

opened plasmids to generate pUbiqP_mCherry-Nup358-N1 (aa 1–2342), -N2 (aa 1–1970), -N3 (aa 1–1498), -N4 (aa 1–1269),- and -N5 (aa 1–830). To create plasmid derivatives for expression of C-terminal fragments, the five pUbiqP_mCherry-Nup358 versions with introduced NheI sites were digested with NheI and NotI, and closed by ligation with a linker obtained by annealing the primers LP028 and LP029, yielding pUbiqP_mCherry-Nup358-C1 (aa 2342–2718), -C2 (aa 1970–2718), -C3 (aa 1498–2718), -C4 (aa 1269–2718), and -C5 (aa 830–2718). The additional constructs pUbiqP_mCherry-Nup358-N6 (aa 1–2484), -N7 (aa 1–2538), and -N8 (aa 1–2698) were generated by double digestion of pUbiqP-mCherry-Nup358-Nhe1 (vector) and the synthetic DNA fragments LP054, LP055 and LP056 (inserts) with NheI and AvrII before ligation.

## Fertility tests

The fertility of *Mst27D*[LL] hemizygous flies was analyzed after crossing *Mst27D*[LL]/CyO and *Df (2L)ade3/CyO, P{w+, Dfd-YFP}* to generate the genotypes *Mst27D*[LL]/*Df(2L)ade3* and *Mst27D*[LL]/CyO, P{w+, Dfd-YFP}* as well as *Df(2L)ade3/CyO*. The fertility of these genotypes was compared to that of *w*[1118] flies. Moreover, for additional comparison, the fertility of *Mst27D*[LL]/*Df(2L)ade3; g-Mst27D-EGFP/+* flies, generated with additional crosses, was analyzed in parallel. To assess male fertility, 10 single males of each genotype were crossed individually with three virgins of *w*[1118] in a vial at 25°C for 4 days and turned over into second vial for 4 days. To determine female fertility, five replicate crosses were set up. In each cross, three virgin females were mated with three *w*[1118] males. The number of adult F1 progeny eclosing in the second vials was counted.

To determine to which extent the *g-Mst27D* transgene variants (*g-Mst27D-mCherry*, *g-Mst27D_CH-mCherry* or *g-Mst27D_CT-mCherry*) improve the low fertility of *Mst27D*[LL] hemizygous males, fertility tests were performed with the genotypes *w\*; Mst27D*[LL]/*Df(2L)ade3; g-Mst27D* transgene variants/ +, as well as with the genotype *Mst27D*[LL]/*Df(2L)ade3*. In addition, for comparison *w*[1118] control flies were analyzed in parallel. Four replicate crosses were set up for each genotype, except for the *w*[1118] control, where only three replicate crosses were started. For each cross, three males were mated with three virgins of *w*[1118] in a vial at 25°C for 3 days, and then turned over into another fresh vial for 3 days. All adult F1 flies that eclosed in the second vials were counted.

To assess the effect of the *Mst27D*[cc] null mutation on fertility, homozygous null mutant males and females, respectively, were mated with *w*[1118] flies and the number of the resulting adult F1 progeny was determined. For comparison, analogous *inter se* crosses of *w*[1118] flies were analyzed. To assess male fertility, 10 single males were crossed individually with three *w*[1118] virgins at 25°C for 2 days, and then turned over into a second vial for 3 days. To determine female fertility, five replicate crosses were set up with three virgins and three *w*[1118] males at 25°C for 2 days, and then turned over into a second vial for 3 days. Adult F1 progeny eclosing from the second vials were counted.

## Testis mass isolation

Testis mass isolation was performed as described [35]. Flies with genotype *w\*;att22A{exum-mCherry}/ CyO; g-Mst27D-EGFP/ TM6B, Tb, Hu* were expanded to start cultures in 25 large bottles (4.7 cm diameter × 9.5 cm high, with fly food filling the bottom 2.5 cm). The resulting flies were distributed into two collection cages (80 × 80 × 80 cm). In each cage, four round plates containing standard fly food (14 cm diameter × 2 cm height) were placed for egg collection. Every 8 hours at 25°C or 16 hours at 18°C, the plates were replaced with four fresh plates. The plates with eggs were cultured at 25°C for 2 days. Afterwards, the food and larvae from the

plates were transferred to fresh food plates and plexiglass puparation cylinders (14 cm diameter × 50 cm height) were inserted on top. After 18 hours incubation at 25°C, early pupae and wandering stage 3$^{rd}$ instar larvae were dislodged from the wall of the cylinders with a silicon spatula and a flow of water. Larvae and early pupae were collected in a first sieve with 700 μM pore size. Afterward, pupae and larvae were re-suspended in Ringer Buffer (182 mM KCl; 46 mM NaCl; 3 mM CaCl2, 10 mM Tris-HCl, pH 7.2) and minced in a metal mill attached to a food processor (KSM125 equipped with KGM, Kitchen Aid). The disrupted tissue pieces were washed out of the mill by a continuous flow of Ringer buffer that was passed through a second sieve with 355 μM pore size and a third sieve with 100 μM pore size. The material in the second sieve was re-loaded into the mill for 2–3 additional rounds of mincing. All material from the third sieve was washed away with Ringer buffer and collected in Falcon tubes for further purification.

Ficoll gradient density centrifugation was used for a first gonad purification step. The gradient was prepared with one bottom layer of 20 ml of 25% Ficoll PM400 and one upper layer of 20ml 12% Ficoll PM400 in Ringer buffer. After settling of the tissue material in the Falcon tubes for 15 minutes, supernatant was discarded and sediments were transferred on the top of the Ficoll gradient. The gradient tubes were centrifuged at 4500 × g for 1 hour at 4°C. Afterwards, the boundary regions between the 25%/12% and 12%/0% gradient phases, as well as the material in the region in between was collected and diluted 1:5 with Ringer buffer. Fluorescence-based particle sorting was performed as an additional purification step as described [89] with the following modifications. After a settling period of 30 minutes, supernatants were discarded, and the remaining sediments were applied to a Biosorter (Union Biometrica; FOCA 1000; standard settings). Depending on particle density, the suspension was further diluted in Ringer solution for optimal sorting. Gating for selection of testes was based on measurements of the following parameters: time of passage through optical detection cell, extinction of excitation light, green and red fluorescence intensity. The quality of testis selection was assessed by inspection of gated samples with a stereomicroscope. Sorted testes were recovered in Ringer Buffer in a 50 ml Falcon tube with Ringer solution. The testes were collected by centrifugation at 4500 × g for 30 seconds. The supernatant was discarded, and the sediment was transferred to a 1.5 ml low binding Eppendorf tube (Eppendorf, #0030108116) and snap frozen in liquid nitrogen after settling of testes and removal of supernatant.

## Affinity purification of EFGP fusion proteins

Affinity purification of Mst27D-EGFP and control EGFP fusion proteins was performed as described [35]. Four replicates, each with approximately 9000 testes, were processed for each protein. Testes were homogenized in 1.5 ml lysis buffer (20 mM Tris-HCl at pH 7.5, 150 mM NaCl, 2 mM MgCl$_2$, 0.1% Nonidet NP-40 [Sigma Aldrich IGEPAL CA-630], 5% glycerol, 1 tablet complete protease inhibitor cocktail [Roche #04693159001]/10 ml, 1 mM DTT and 50 U/ml Benzonase) using a Potter-Elvehjem homogenizer with a glass pestle. The lysate was centrifuged at 16000 × g for 15 minutes at 4°C. The supernatant was transferred to a new low binding Eppendorf tube and the centrifugation step was repeated. Thereafter, the supernatant was added to 50 μl μMACS anti-GFP beads (Miltenyi Biotec, Solothurn, Switzerland, #130-091-125), and incubated for one hour on a rotating wheel at 4°C. The suspension was transferred into a pre-equilibrated μColumn (Miltenyi Biotec, #130-042-701). Exploiting the magnetic field provided by the magnetic stand (Miltenyi Biotec, #130-042-602 and #130-042-303), the column was washed four times with 200 μl lysis buffer and once with Tris-HCl (pH 7.5). All steps were performed at 4°C.

The PreOmics iST kit (PreOmics GmbH, Planegg/Martinsried, Germany, #P.O.00001) was used for preparation of samples for MS analyses. At first, 25 μl of LYSE buffer (PreOmics iST

kit) were added to the column. The column was then removed from the magnetic stand and placed on top of a 1.5 ml Eppendorf tube. Another 50 μl of LYSE buffer were added to the column to elute the beads and proteins onto the cartridge (PreOmics iST kit). Reduction and alkylation were performed at 60°C for 10 minutes in a shaker at 1000 rounds per minute (rpm). Digestion with Trypsin plus Lys-C and the following purification steps were performed according to the manufacturer's instructions.

## Liquid chromatography, mass spectrometry and sequence comparison

Mass spectrometry was performed as described [35]. Samples were injected by an Easy-nLC 1000 system (Thermo Scientific) and separated on an EasySpray-column (75 μm x 500 mm) packed with C18 material (PepMap, C18, 100 Å, 2 μm, Thermo Scientific). The column was equilibrated with 100% solvent A (0.1% formic acid (FA) in water). Peptides were eluted using the following gradient of solvent B (0.1% FA in acetonitrile): 5% B for 2 minutes; 5–25% B in 60 minutes; 25–35% B in 10 minutes; 35–95% B in 5 minutes at a flow rate of 0.3 μl/minutes. High accuracy mass spectra were acquired with an Orbitrap Fusion (Thermo Scientific) that was operated in data dependent acquisition mode. All precursor signals were recorded in the Orbitrap using quadrupole transmission in the mass range of 300–1500 m/z. Spectra were recorded with a resolution of 120'000 at 200 m/z, a target value of 5E5 and the maximum cycle time was set to 3 seconds. Data dependent MS/MS were recorded in the linear ion trap using quadrupole isolation with a window of 1.6 Da and HCD fragmentation with 30% fragmentation energy. The ion trap was operated in rapid scan mode with a target value of 8E3 and a maximum injection time of 80 ms. Precursor signals were selected for fragmentation with a charge state from +2 to +7 and a signal intensity of at least 5E3. A dynamic exclusion list was used for 25 seconds and maximum parallelizing ion injections was activated. After data collection the peak lists were generated using Proteome Discoverer 2.1 (Thermo Scientific).

All MS/MS data were analyzed using Mascot 2.4 (Matrix Science, London, UK). MS data were searched against a decoyed database from UniProt (organism:7227, date:20131216) concatenated with an in-house build contaminant database. Precursor ion mass tolerance was set to 10 ppm and the fragment ion mass tolerance was set to 0.6 Da. The following search parameters were used: trypsin digestion (two missed cleavages allowed), fixed modifications of carbamidomethyl labelled cysteine, and as variable modification oxidation of methionine, transformation of N-terminal glutamine to pyroglutamic acid (Gln→pyro-Glu), and deamidation of asparagine and glutamine. Data was filtered using Scaffold (v4.10.0) with the settings Protein threshold 95%, Minimal number of peptides 2 and Peptide Threshold DTs. The number of the peptides displayed in Fig 1D corresponds to the rounded Quantitative Value (Normalized Weighted Spectra) provided by the scaffold software.

For the comparison of amino acid sequence similarity between human EB1 and the *D. melanogaster* EB1 protein family members (Fig 1C), sequences were retrieved from UniProt (Q15691 for human EB1/MAPRE1) and FlyBase (Eb1-PA, CG32371-PA, CG18190-PB, CG15306-PA, CG2955-PA and Mst27D-PA). After aligning the sequences using Clustal Omega with default settings provided by the EMBL-EBI online services, the cladogram of the guide tree was downloaded for incorporation into Fig 1C.

## Cell culture, transfection and microscopic analyses

S2R+ cells were cultured as described [90]. S2R+ cells were grown in Schneider's medium (Gibco, #21720) supplemented with 10% fetal bovine serum (Gibco, #10500–064) and 1% Penicillin-Streptomycin (Gibco, #15140) at 25°C. For transfection, S2R+ cells were seeded in a 25-cm$^2$ flask (Greiner Bio-ONE, #690160) with $2.6 \times 10^6$ cells in 4 ml complete medium, or in

a 35 mm glass bottom dish (MatTek Corporation, #P35G-1.5-14-C) with $6 \times 10^5$ cells in 2 ml, or in a single well of 24-well plate (with a sterilized coverslip at bottom) with $1.5 \times 10^5$ cells in 0.5 ml medium. After allowing cells to adhere for 1 hour at 25°C, they were transfected. The transfection mixture for a 25-cm$^2$ flask contained 2 μg of plasmids, 8 μl FuGENE HD (Promega, #E2311) and Schneider's medium (not supplemented) added to a total volume of 200 μl; in case of a 35 mm glass bottom dish 1 μg of plasmid, 4 μl FuGENE HD and Schneider's medium to 100 μl total volume, and in case of a single well of 24-well plate 0.25 μg plasmids, 1 μl FuGENE HD and Schneider's medium to 25 μl total volume. The transfection mixture was incubated at room temperature for 15 minutes and added dropwise to the cells, followed by an incubation at 25°C for 48 hours. For the induction of expression from pMT plasmid constructs under control of the *Metallothionein A* (*MtnA*) promoter, CuSO$_4$ was added (500 μM final concentration) 24 hours after transfection followed by additional 24 hours of incubation. Cells were then either fixed or analyzed by live imaging or harvested for immunoprecipitation experiments. For the establishment of stable cell lines, medium was replaced 48 hours after transfection with fresh medium containing blasticidin S (25 μg/ml final concentration). Selection was continued for 10–16 days.

To detect expression of fluorescent fusion proteins in S2R+ cells grown on coverslip or in a glass bottom dish, cells were fixed with either 4% formaldehyde in phosphate-buffered saline (PBS) (137 mM NaCl, 2.7 mM KCl, 1.47 mM KH$_2$PO$_4$, 6.46 mM Na$_2$HPO$_4$, pH 7.4) at room temperature or cold methanol at -20°C for 10 minutes. After a brief wash with PBS, cells were permeabilized with 0.5% Triton-X-100 in PBS for 3 minutes. After two washes with 0.1% Tween-20 in PBS, DNA was stained with Hoechst 33258 (1 μg/ml) for 5 minutes. After two washes with PBS, around 10–20 μl mounting medium (70% glycerin, 50 mM Tris-HCl pH 8.5, 10 mg/ml propyl gallate, 0.5 mg/ml phenylendiamine) as described above was added onto a slide, and the coverslip with the cells was inverted and lowered onto the mounting medium. In case of glass bottom dishes, mounting medium was added and covered with a coverslip.

For immunofluorescent staining of MTs, cells were fixed with cold methanol. Mouse monoclonal antibody DM1A anti-α-tubulin (Sigma, #T9026; 1:8000) was used as primary antibody and Cy5-conjugated goat anti-mouse antibody (Invitrogen, #A10524; 1:500) as secondary antibody.

The slides were imaged using an inverted wide-field fluorescence microscope (Zeiss Cell Observer HS) with 40×/1.3, 63×/1.4 or 100×/1.4 oil immersion objectives. To analyze the expression of fluorescent fusion proteins by live imaging, preparations in glass bottom dishes were mounted on a spinning disc confocal microscope (VisiScope with a Yokogawa CSU-X1 unit combined with an Olympus IX83 inverted stand and a Photometrics evolve EM 512 EMCCD camera, equipped for red/green dual channel fluorescence observation; Visitron systems, Puchheim, Germany) with a 40×/1.3 or 60×/1.42 oil immersion objective at 25°C.

### RNAi with cultured cells

The template for *Nup358* dsRNA production was amplified from $w^1$ genomic DNA with the primer pair LP045/LP046. The plasmid pUR291 was used for the amplification of the template for *lacZ* control dsRNA production with the primer pair CL298/CL299. PCR products were purified with a gel extraction kit (QIAGEN, #28706). The purified products were used to template *in vitro* transcription using the MEGAscript T7 Transcription Kit (Thermo, #AM1334). The dsRNA samples were purified with the MEGAclear Transcription Clean-Up Kit (Thermo, #AM1908).

Depletion with *Nup358* or *lacZ* dsRNA was performed with S2R+ cells stably expressing mCherry-Nup358 or Mst27D_CT-EGFP as described [91]. Cells were harvested and re-

suspended in Schneider's medium (without supplements) at a density of $1 \times 10^6$ cells/ml. 1.5 ml cell suspension was mixed with 15 μg dsRNA and seeded into a 35 mm glass bottom dish. After the incubation of cells with dsRNA for 45 minutes, 3 ml complete medium was added. Cells were incubated at 25°C for 3 days before microscopic analysis.

## Immunoprecipitation and immunoblotting

For co-immunoprecipitation experiments, S2R+ cells were collected by a cell scraper from two 25-cm² flasks and centrifuged at $580 \times g$ for five minutes. All further steps were performed on ice or at 4°C. Cells were re-suspended with 1 ml PBS and transferred to a 1.5 ml low binding Eppendorf tube. This tube type was also used for all additional steps. Cells were sedimented by centrifugation at $600 \times g$ for 5 minutes at 4°C. Cells were re-suspended in 600 μl lysis buffer (LB) and incubated on ice for one hour. Insoluble material was removed by centrifugation at $16100 \times g$ for 15 minutes. A small fraction (10%) of the supernatant was reserved for later analysis by immunoblotting (input samples). The rest of the supernatant (540 μl) was incubated with 25 μl anti-RFP beads (ChromoTek, RFP-Trap agarose, rta-20) for 1 hour on a rotating wheel at 15 rpm. Beads were collected by centrifugation at $2500 \times g$ for two minutes and washed 3 times with 500 μl LB by repeating the centrifugation steps. For the final wash, beads were transferred into a fresh tube. After centrifugation, beads were re-suspended in 80 μl 3x Lämmli Buffer (62.5 mM Tris-HCl, 10% glycerol, 5% β-mercaptoethanol, 3% SDS, 0.01% Bromophenol Blue) and boiled for 10 minutes at 96°C. After two minutes of incubation on ice, samples were centrifuged to sediment the beads and the supernatant was distributed into aliquots for latter analysis by immunoblotting (IP samples). All samples were snap frozen in liquid nitrogen and stored at -80°C until immunoblotting analysis.

For immunblotting, samples were resolved by sodium dodecyl sulfate-polyacrylamide gel electrophoresis (SDS-PAGE). Input and IP samples were heated to 96°C for 5 minutes and centrifuged at $13100 \times g$ for 5 minutes before loading the supernatant onto the SDS-PAGE gels with 6.5% or 10% polyacrylamide. Electrophoresis was performed at 80 V during about 2 hours. Prestained Protein Ladder (Thermo Fisher, #26619) was used as molecular weight marker. Thereafter, proteins were transferred from the gel to Hybond ECL nitrocellulose membranes (Amersham Protran 0.45 μm, #10600002) by tank blotting overnight using 30 V. Membranes were stained with Ponceau S solution (Sigma, P7170) to assess loading quantities and electro-transfer efficiencies. After brief washing with PBS, membranes were blocked by incubation in 5% dry milk (w/v) in PBS with 0.02% NaN$_3$ for 30 minutes at room temperature. The primary antibody was added into this solution for an incubation for 2 hours at room temperature or overnight at 4°C. Three washes with 5% dry milk (w/v) in PBS were performed before incubation with the secondary antibody, which was applied in 5% dry milk (w/v) in PBS at room temperature for 1 hour in a dark environment. After 3 washes with 5% dry milk (w/v) in PBS and 2 washes with PBS, 0.1% Tween-20, signals were detected with ECL reagents (WesternBright ECL, Advansta, #K-12045-D50) in an Amersham Imager 600 system.

Mouse monoclonal antibody anti-RFP (ChromoTek, #6g6) was used at 1:1000 as primary antibody and HRP-conjugated AffiniPure goat anti-mouse IgG polyclonal antibody (Jackson ImmunoResearch, 115-035-003) was used at 1:1000 as secondary antibody. This secondary antibody at the same dilution was also used to detect mouse anti-Lamin ADL67.10 (Developmental Studies Hybridoma Bank) used as primary antibody at 1:200. In addition, rabbit polyclonal antibodies anti-GFP diluted 1:1000 (ChromoTek, #pabg1) or 1:2000 (Torrey Pines Biolabs, #TP401) were used as primary antibodies and HRP-conjugated AffiniPure goat anti-rabbit IgG polyclonal antibody (Jackson ImmunoResearch, #111-035-003) at 1:1000 as secondary antibody. This secondary antibody at the same dilution was also used to detect affinity-

purified rabbit anti-Mst27D N1, I1 or C1 (Moravian Biotechnology, Brno, Czech Republic) used as primary antibodies at 1:10000, 1:3000 and 1:2000, respectively.

## Fixed preparations of testes

For whole mount preparations, testis dissection was performed in testis buffer (183 mM KCl, 47 mM NaCl, 10 mM Tris-HCl, pH 6.8). For each slide, 10–20 testes were dissected. For permeabilization, testes were incubated twice in PBST-DOC (PBS containing 0.3% Triton-X100 and 0.3% sodium deoxycholate) for 15 minutes each. Testes were fixed in PBS containing 4% formaldehyde and 0.1% Triton X-100 for 20 minutes on a rotating wheel. After a rinse in PBS containing 0.1% Triton X-100 (PBST), blocking solution (PBST containing 5% fetal bovine serum) was added for 30 minutes. Thereafter, primary antibody was added into fresh blocking solution for incubation overnight at 4˚C, or for two hours at room temperature on a rotating wheel. Testes were then washed four times in PBST for 15 minutes each. Secondary antibody was added in blocking solution for one hour at room temperature. After four washes with PBST, testes were incubated for 10 minutes in PBST containing 1 µg/ml Hoechst 33258. After three additional washes with PBS, testes were transferred into 10 µl of mounting medium on a slide before adding a cover slip. For staining of F-actin (S7A and S11 Figs), Alexa647-conjugated Phalloidin (Invitrogen, #A22287) was applied at 1:500 for 30 minutes at room temperature after the incubation with secondary antibody and before DNA staining. To determine the expression pattern and subcellular localization of Mst27D, affinity-purified rabbit antibodies N1, I1 or C1 against Mst27D were used as primary antibodies at a dilution of 1:1000. Alexa488-conjugated goat anti-rabbit IgG (Invitrogen, #A11008) was used as secondary antibody at 1:500 (Fig 5). Beyond the specific signals illustrated in Fig 5, the different anti-Mst27D antibodies resulted in signals that did not represent binding to Mst27D, as revealed by staining of *Mst27D* null mutants, to variable degrees at various locations within the testes. Specific anti-Mst27D signals were stronger in whole mount preparations compared to squash preparations of testes.

Testis squash preparations were made and stained essentially as described previously [92], according to protocol 3.3.2, using the mounting medium described above. Before fixation and immunofluorescent labeling, late larval and pupal testes were dissected in testis buffer. Testes were transferred to a drop of testis buffer (20 µl) on a poly-L-lysine-coated slide and opened with a tungsten needle. Cysts were dispersed and flattened by placing a 22 mm x 22 mm cover slip onto the drop for two minutes before freezing the preparation in liquid nitrogen. Thereafter, the cover slip was removed with a surgical scalpel and the slide was immersed into ethanol for 10 minutes at -20˚C. The region with the squashed testes was fixed by adding 0.4 ml of 4% formaldehyde in PBS for 10 minutes within an area marked by a hydrophobic marker pen. Permeabilization was done twice in PBST-DOC for 15 minutes each. After a rinse in PBST, blocking solution (PBST containing 5% fetal bovine serum) was added for 30 minutes. Primary antibody was added in the marked area and the slide was then incubated in a humid chamber overnight at 4˚C, or for two hours at room temperature. After a quick rinse with PBST, slides were incubated in PBST for 15 minutes four times. The secondary antibody was added for one-hour at room temperature. After washing with PBST, Hoechst 33258 (1 µg/ml in PBS) was applied for DNA staining for five minutes followed by three washes with PBS. Finally, the preparation was mounted with a drop of mounting medium and a cover slip.

Images were acquired using a Zeiss Cell Observer HS wide-field microscope. Z-stacks were acquired with 40×/1.3, 63×/1.35 and 100×/1.42 oil immersion objectives for analysis at high resolution. Spacing between focal planes was 280 nm. To visualize the structure of the investment cones (S7A and S11 Figs), images were acquired using an Olympus FluoView 1000 laser-

scanning confocal microscope using a 40x/1.3 oil or 60×/1.42 objective and a Kalman filter of 3.

## Live imaging of testes

Time-lapse imaging of cysts released from testes was performed as described [93]. Depending on the stages to be analyzed, testes were dissected from pre-pupae or 1-day old adult males in complete Schneider's Medium in a depression well. Pre-pupal testes were separated from fat body remnants and transferred into 45 µl of medium in a 35 mm glass bottom dish. In the case of adult testes, 150 µl of medium was used. Each testis was ruptured with tungsten needles to release and separate cysts. Thereafter, 15 µl of 1% w/v methylcellulose (Sigma, #M0387) in complete Schneider's medium was applied to pre-pupal testes preparations and 50 µl to adult testes preparations to settle down the floating samples. A filter paper soaked with water was then placed along the inner wall of the dishes. The lid was closed and sealed with parafilm before mounting the dish on the stage of a spinning disc confocal microscope. Imaging was performed at 25˚C in a room with precise temperature control. Images were acquired with a 100×/1.4 or 60×/1.42 oil immersion objective, and the acquired z-stacks comprised 30 to 40 focal planes spaced by 500 nm usually recorded at 2 min time intervals except for S6B Fig (45 sec intervals), Figs 7A–7C, 7E and S10A (90 sec intervals).

For live tubulin staining in S2R+ derived cell lines, SPY555-tubulin [94] (Spirochrome, obtained from Lubio Science, #SC203) was added to the preparations at a final dilution of 1:1000. Demecolcine (Sigma, #D6165) was dissolved in dimethyl sulfoxide (10 mM) before further dilution of this stock solution to a final concentration of 10 µM in complete Schneider's Medium. After demecolcine addition, samples were immediately mounted on the microscope and imaging was started about 20 minutes after demecolcine addition.

For the FRAP and Dendra2 photoconversion experiments, an Olympus FluoView 1000 laser-scanning confocal microscope was used. For FRAP with Eb1-tdGFP or EGFP-alpha-Tub84B, testes were isolated from 1-day-old males with a genotype that was either $w^*$; $EB1$-$tdGFP(II)$ or $w^*$; $P\{w+, Ubq11$-$EGFP$-$alphaTub84B\}$; $P\{w+, gMst27D$-$mCherry\}$ $III.7$. Image stacks consisting of three z-sections spaced by 500 nm were acquired. A circular ROI was selected to cover around a quarter of the medial region of the dense complex in spermatids with nuclei that were already considerably elongated. After acquisition of five initial image stacks, photobleaching was completed during one second with 405 nm light using 100% intensity. Thereafter, 20 additional image stacks at three seconds intervals were acquired in case of Eb1-tdGFP and 50 at 20 seconds intervals in case of EGFP-alphaTub84B. In the Mst27D-Dendra2 photo-conversion experiments, 20 image stacks with 10 z-sections spaced by 500 nm were acquired at three minutes interval. A circular ROI was selected to cover around a quarter of the NE rim and photo-conversion was started manually during acquisition at the second time point using 405 nm light with 1.0% laser intensity during one second.

## Image processing and analysis

Maximum intensity projections were generated using the software Fiji or ZEN (version 2.3) (Carl Zeiss AG, Oberkochen, Germany) for wide-field images and IMARIS (Bitplane; versions 8.4.0, 9.2.0, 9.7.2) for most of the spinning disk and laser scanning confocal images. For export of projections as movies or as still frame images after live imaging the option of interpolated image display was activated when using IMARIS. Moreover, sample movements were corrected using drift correction with IMARIS before exporting the S3–S3 Movies.

For quantification of the effects of Nup358 degradation by deGradFP in S6 spermatocytes (Fig 4), single optical sections through the equatorial region of the nucleus were used for

quantification of signal intensities with Fiji. Five ROIs (cytoplasm left, NE left, nuclear center, NE right, cytoplasm right) were chosen as illustrated (Fig 4D). For ROI selection, a line with a width set to 16 was drawn through the center of the nucleus. Signal intensities along this line were inspected to identify the position where the line crosses the NE on the left and on the right of the nuclear center based on local EGFP-Nup358 intensity maxima. The values of five positions along the line centered around the position of maximal intensity value were averaged for determination of signal intensities at the NE (NE_left and NE_right, respectively). For background correction of these signal intensities, the intensities at the five central positions along the line (nuclear center) were averaged and subtracted. For estimation of the signal intensities in the cytoplasm, the average value of a window of five positions along the line shifted away from the NE by around 20 pixels was determined, and for background correction of these cytoplasmic signal intensities we also subtracted the average of the intensity values in the five central most positions (nuclear center). For quantification of the effects of Nup358 depletion by RNAi (S3C Fig), an analogous approach was used. Single optical sections through the equatorial region of S2R+ cells expressing Mst27D_CT-EGFP were quantified. Integrated intensity of the Mst27D_CT-EGFP signals was measured in the ROIs selected in the cytoplasm, at the NE and in the central region of the nucleus. For each cell, the intensities in the two cytoplasmic ROIs were averaged, as well as those in the two ROIs on the NE. The intensities observed within the ROI in center of the nucleus were considered to reflect background, which was subtracted from the intensities detected in the cytoplasm and at the NE.

For the quantification of Mst27D-EGFP signal intensities (S8 Fig) resulting with *g-Mst27D-EGFP* and *g-Mst27D-EGFP-endo3'*, respectively, maximum intensity projections of image stacks covering the entire cluster of nuclei of a given cyst were analyzed with Fiji. In the case of cysts with early spermatids, an oval ROI was selected manually around a given nucleus. In the case of late spermatids, ROIs around the entire cluster of nuclei were selected manually using the freehand selection tool. The integrated intensity within this ROI was measured. Thereafter, the size of the ROI was dilated by the width of 3 pixels. The integrated intensity measured within this larger ROI was used for background correction using the following formulas: background intensity per pixel = (RawIntDen in large ROI–RawIntDen in small ROI/ (area of large ROI–area of small ROI); background corrected intensity = RawIntDen of small ROI—(background intensity per pixel × area of small ROI).

For quantification of signal intensities obtained in the FRAP or photo-conversion experiments, the ROI used for bleaching and photo-conversion, respectively, was also used for signal quantification at time points before and after bleaching/photo-conversion. However, the position of the ROI was manually adjusted at each time point to compensate sample movements. Sum slice projections were made with Fiji for the quantification of the Mst27D-Dendra2 signal intensities (S4 Fig).

Data obtained after quantification with Fiji or IMARIS was exported and further processed using Microsoft Excel or GraphPad Prism (8.0.1) for plotting. P values were calculated using the Wilcoxon rank sum test (* = $p < .05$; ** = $p < .01$; *** = $p < .001$). Adobe Photoshop, Illustrator or Inkscape were used for assembly of figures.

## Supporting information

**S1 Fig. Subcellular localization of mCherry-Nup358 fragments.** S2R+ cells transiently expressing the indicated Nup358 fragments tagged with mCherry at the N terminus (see Fig 2B for additional information) were fixed for microscopic analysis. While expression levels varied substantially from cell to cell, the displayed images are from cells with relatively weak mCherry signals, in which NE enrichment was most clearly detectable if present. Fragments

N1 –N4 were strongly enriched at the NE, comparable to full length mCherry-Nup358. In contrast, N5 and C1 –C4 were at most very weakly enriched at the NE. C5, which includes the OE but not the NTD, was usually in sheet-like aggregates that were perinuclear in cells with less extensive aggregates. Transient expression of C1 –C5 was also accompanied by an accumulation of apoptotic S2R+ cells that was not observed with N1 –N5. Scale bar = 5 μm.
(PDF)

**S2 Fig. High levels of Mst27D expression induces MT bundling.** (**A**) S2R+ cell lines expressing either Mst27D-EGFP or Mst27D-mCherry were analyzed by live imaging. Still frames display cells with prominent intracellular cables that were observed in cells characterized by high expression levels. Scale bar = 5 μm. (**B**) S2R+ cell lines expressing either Mst27D-EGFP or Mst27D_CH-EGFP were incubated with the MT live stain SPY555-tubulin before live imaging. Cells with high levels of Mst27D-EGFP display prominent MT cables in contrast to cells with high levels of Mst27D_CH-EGFP. Scale bar = 10 μm. (**C**) S2R+ cell lines expressing Mst27D_CH-Eb1_CT-EGFP, the EGFP tagged chimeric protein with the CH domain of Mst27D and the CT region of *D. melanogaster* Eb1, were fixed and double labeled with anti-α-tubulin. A weakly (top) and a strongly (bottom) expressing cell are shown, revealing the bundling of MTs into prominent intracellular cables at high expression levels. Scale bar = 5 μm.
(PDF)

**S3 Fig. Subcellular localization in S2R+ cells indicates interaction of Mst27D and Nup358.** (**A**) S2R+ cells transiently co-expressing mCherry-Nup358 and either Mst27D-EGFP or Mst27D_CH-Eb1_CT-EGFP were fixed for microscopic analysis. Representative cells with strong MT cables are displayed. While mCherry-Nup358 is recruited to the MT cables induced by Mst27D-EGFP at high levels of expression, mCherry-Nup358 is not recruited to MT cables induced by high levels of Mst27D_CH-Eb1_CT-EGFP expression, as expected because the chimeric protein lacks the CT region of Mst27D that mediates binding to Nup358. (**B,C**) Effects of Nup358 depletion by RNAi in S2R+ cells. (**B**) Incubation of S2R+ cells with *Nup358* dsRNA results in the expected depletion of Nup358. S2R+ cells stably expressing mCherry-Nup358 were treated with either *Nup358* or *lacZ* dsRNA for control. DNA was labeled after fixation. (**C**) Nup358 depletion abolishes the enrichment of Mst27D_CT-EGFP at the NE. S2R+ cells stably expressing Mst27D_CT-EGFP were treated with either *Nup358* or *lacZ* dsRNA before live imaging of EGFP signals. Single optical sections through representative cells are displayed. The weak enrichment of Mst27D_CT-EGFP on the nuclear rim and more prominently on putative annulate lamellae (arrowheads) that is evident in *lacZ* dsRNA treated controls is absent after treatment with *Nup358* dsRNA. EGFP signal intensities associated with the NE and within the cytoplasm were quantified. Swarm plots display values from individual cells; mean and s.d. are indicated as well. n = 54 (*lacZ* dsRNA) and 76 (*Nup358* dsRNA). Nup358 depletion also reduced the overall level of Mst27D_CT-EGFP, suggesting that Nup358 is required for normal expression of Mst27D_CT-EGFP, perhaps reflecting a Nup358 requirement for normal translation, as recently shown with human HCT116 cells [37]. Scale bars = 5 μm (A), 20 μm (B) and 3 μm (C).
(PDF)

**S4 Fig. Dynamics of the NE association of Mst27D.** (**A**) Cysts with S6 spermatocytes expressing *g-Mst27D-Dendra2* were analyzed by time-lapse imaging. The green initial fluorescence of Mst27D-Dendra2 in a region containing part of the NE (dashed circle) was photoconverted in one of the spermatocytes. Photoconversion was performed concomitant with acquisition of the z-stack at time 0. Redistribution of photoconverted red fluorescent Mst27D-Dendra2 was analyzed over time. Still frames display the green and red fluorescent Mst27D-Dendra2 signals,

as well as their merge, as maximum intensity projections of 10 optical sections with 500 nm spacing. Scale bar = 5 μm. (**B**) For analysis of the dynamics of dispersal of photoconverted red fluorescent Mst27D-Dendra2, signal intensities in the red channel were quantified within four equal sized regions of interest (ROIs) (1–4). Average signal intensities (+/- s.d.) over time in the four ROIs are displayed (n = 5 S6 spermatocytes from distinct cysts).
(PDF)

**S5 Fig. Transgenes for deGradFP in spermatocytes induce EGFP-Nup358 degradation.** (**A**) Spermatocyte-specific degradation induced by the transgenes *betaTub85DP-deGrad* and *exumP-deGrad*. The functionality of these *deGrad* transgenes was evaluated in *g-cid-EGFP* testes, which express the centromere protein Cid/Cenp-A-EGFP, an EGFP fusion protein previously shown to be degraded efficiently by deGradFP [95]. Apical testis regions from whole mount preparations are displayed. While centromeric Cid-EGFP dot signals are present in all cells in the control (no *deGrad*), these signals are lost from spermatocytes distal from the dashed lines in *g-cid-EGFP* testes with *betaTub85DP-deGrad* or *exumP-deGrad*. (**B**) Spermatocyte-specific degradation of EGFP-Nup358 by *exumP-deGrad* in early pupal testes. Whole mount preparations were fixed and labeled with a DNA stain. The anterior regions with the initial stages of spermatogenesis and the posterior regions with late spermatocytes of representative -*deGrad* and +*deGrad* early pupal testes are shown with identical settings for imaging and display. A subregion (dashed rectangle) is shown with enhanced signals in the inset to reveal residual EGFP-Nup358 in late spermatocytes of +*deGrad* testes. Scale bars = 20 μm
(PDF)

**S6 Fig. Localization of Mst27D and Nups during the meiotic division and NE polarization in early round spermatids.** (**A**) Time-lapse imaging of cells expressing Mst27D-mCherry and EGFP-Nup358. Still frames display a spermatocyte from a first movie, documenting progression through meiosis I (M I) and meiosis II (M II), and an early post-meiotic spermatid from a second movie, documenting NE polarization. Time points (h:min) are indicated. In the first movie: NEBD I (0:00), metaphase I (0:32), anaphase I (0:48), telophase I (1:02), interkinesis (1:22, 1:38), NEBD II (2:32), metaphase II (3:12), anaphase II (3:30), telophase II (4:00) and post-meiotic interphase (4:46). In the second movie, t = 0:00 corresponds to early post-meiotic interphase, and the rapidly weaking Mst27D-mCherry enrichment on the centrosome is indicated (arrowhead). (**B**) Time-lapse imaging of a spermatocyte expressing Nup58-EGFP and histone H2Av-mRFP (His2Av-mRFP) during progression through meiosis I (M I) and meiosis II (M II). Time points (h:min) are indicated with t = 0 at the onset of NEBD I. Scale bars = 3 μm.
(PDF)

**S7 Fig. Stage and dynamics of the NPC-NE shedding process.** (**A**) Stage of NPC-NE shedding. Whole mount preparations of *g-Nup58-EGFP* testes were stained with fluorescent phalloidin to detect F-actin and with a DNA stain. Maximum intensity projections of testes regions with clustered spermatid nuclei of cysts during the stages of NPC-NE shedding and F-actin cone formation are displayed. Sperm head clusters of increasing ages are indicated with numbers: before shedding (1), early after shedding (2), and later stages (3–5). F-actin is not just prominent in the individualization cones but also in the muscle sheet around the testis tube (asterisk). (**B**-**D**) Dynamics of NPC-NE shedding. Spermatid cysts expressing *g-Nup58-EGFP* and *g-ProtB-DsRed* were analyzed by time-lapse imaging. (**B**) Still frames displaying single optical sections through the region with clustered elongated spermatid nuclei at the indicated time points (min). (**C**) Still frames displaying a single spermatid nucleus at the indicated time points (min). The original images acquired with spinning disk confocal microscopy were

deconvolved before maximum intensity projection. (**D**) A weak Nup58-EGFP signal remains at the base of the elongated nuclei after completion of NPC-NE shedding (arrows). (**E**) The weak Nup58-EGFP signal at the base of the elongated nuclei is also detectable in mature sperm released from testis by squash preparation labeled with a DNA stain. The faint green dot signal is detected in *g-Nup58-EGFP* but not in *w^1118* control sperm. Scale bars = 5 μm (A,B,D,E) and 3 μm (C).
(PDF)

**S8 Fig. *Mst27D* 3' sequences prevent premature Mst27D protein accumulation in spermatocytes.** (**A**) Comparison of the pattern of Mst27D-EGFP accumulation driven by *g-Mst27D-EGFP-endo3'* (endo3'), a transgene with downstream sequences identical to those at the endogenous *Mst27D* locus, and by *g-Mst27D-EGFP* (SV40), a transgene with SV40 terminator sequences replacing the endogenous *Mst27D* 3' sequences. Whole mount preparations are displayed on the left (scale bar = 100 μm). Accumulation in spermatocytes (arrowheads) results with *g-Mst27D-EGFP* but not with *g-Mst27D-EGFP-endo3'*. High magnification views of single optical sections through nuclei at the indicated stages are displayed on the right (scale bars = 5 μm). Mst27D-EGFP is detectable in late spermatocyte nuclei in *g-Mst27D-EGFP* (SV40) testis but not in *g-Mst27D-EGFP-endo3'* testis (dashed circle). (**B**) Quantification of Mst27D-EGFP signal intensity associated with the nuclei in spermatids at the start of nuclear elongation (left) and after completion of nuclear elongation (right) in the indicated genotypes. Individual nuclei and entire nuclear clusters were quantified at the early and late stage, respectively. Individual measurements and means (+/- s.d.) are displayed; n = 388 (SV40 early), 301 (endo3' early), 31 (SV40 late) and 55 (endo3' late).
(PDF)

**S9 Fig. Localization and function of the proteins expressed by *g-Mst27D-mCherry*, *g-Mst27D_CH-mCherry* and *g-Mst27D_CT-mCherry* in males with and without endogenous *Mst27D* function.** (**A**) Expression pattern and localization of Mst27D_CT-mCherry and Mst27D_CH-mCherry in testes with endogenous Mst27D function. Whole-mount preparations of testes isolated from transgenic males with *g-Mst27D_CT-mCherry* or *g-Mst27D_CH-mCherry* were fixed and labeled with a DNA stain. The apical regions of testes (left), as well as high magnification views of spermatocytes at the S5 stage (S5) and during prometaphase I (M I), as well as of spermatids early after NE polarization (early) and at the canoe stage (mid) are displayed. While Mst27D_CH-mCherry has a diffuse distribution, Mst27D_CT-mCherry localization corresponds to that of full length Mst27D-mCherry except during the meiotic stages, where the former displayed some enrichment also on the spindle envelope, while the latter is far more prominent on spindle MTs (S6 Fig). (**B-D**) Rescue of the nuclear elongation defect in *Mst27D* mutant spermatids by *g-Mst27D-mCherry* (B) but not by *g-Mst27D_CT-mCherry* (C) and *g-Mst27D_CH-mCherry* (D). The indicated transgenes were crossed into the *Mst27D^LL/ Df(2L)ade3* mutant background. Whole mount testes preparations were labeled with a DNA stain. Single optical sections display late spermatid nuclei at high magnification during the indicated stages. Scale bars = 20 μm for left column in (A) and 5 μm for all other images.
(PDF)

**S10 Fig. MT bundle formation and dynamics in dense complex during nuclear elongation in spermatids.** (**A**) Spermatid cysts expressing EGFP-β-tubulin were analyzed by time-lapse imaging to determine the temporal dynamics of cyst polarization, DC formation and nuclear elongation. Still frames (maximum intensity project of 30 optical sections with 500 nm spacing) from three distinct concatenated movies are displayed with time indicated (h:min; 0:00–

2:00 from S1 Movie, 3:40–10:51 from S2 Movie, and 12:13–19:18 from S3 Movie). Time point t = 0 corresponds to end of M II. High magnification views of regions with basal body and DC-MTs for each time point are shown in bottom panel. (**B**) Effects of demecolcine on DC-MTs and nuclear elongation. Demecolcine was added before the start of time-lapse imaging of a spermatid cyst expressing EGFP-α-tubulin. A complete cyst (top) and high magnification views of a tracked spermatid nucleus (bottom) are displayed with time (h:min) indicated. (**C**) MT dynamics in the DC. Spermatids expressing either Eb1-tdGFP (top) or EGFP-α-tubulin (bottom) were used for fluorescence recovery after photobleaching (FRAP) experiments. Signals within in a central region of the DC-MTs were bleached and signal recovery was monitored at intervals of 3 seconds in case of Eb1-tdGFP and 20 seconds in case of EGFP-α-tubulin. Still frames of maximum intensity projections (three optical z-sections with 0.5 μm spacing) illustrate recovery at the indicated time points (min:sec). The diagram (bottom) represents extent of signal recovery over time. Average signal intensity before and immediately after bleaching were set to 100% and 0%, respectively. Mean values (+/- s.d.) are displayed; n = 6 cysts (Eb1-tdGFP) and 3 cysts (EGFP-α-tubulin). Scale bars = 10 μm (A and B, top), 4 μm (B, bottom) and 3 μm (C).
(PDF)

**S11 Fig. Incomplete shedding of the NPC-NE from *Mst27D* mutant spermatid nuclei.** Testes expressing Nup58-EGFP in a background with *Mst27D* function (*Mst27*⁺) or without (*Mst27D*⁻) were used for whole mount preparations that were labeled with a DNA stain and with fluorescent phalloidin to reveal F-actin. Residual Nup58-EGFP in the NE of nuclei in late *Mst27D* mutant spermatids is indicated (arrows). Scale bar = 10 μm.
(PDF)

**S12 Fig. Independent localization and function of Mst27D and Spag4.** (**A**) Spag 4 is not required for normal Mst27D-EGFP localization in spermatids. Testes expressing *g-Mst27D-EGFP* in a background with *spag4*⁺ function (*spag4*⁺) or in *spag4*¹/*spag4*⁶ mutants (*spag4*⁻) were used for whole mount preparations that were labeled with a DNA stain. Single optical sections display regions from the clustered spermatid nuclei at high magnification. (**B**) Mst27D is not required for normal Spag4-EGFP localization in spermatids. Testes expressing *g-spag4-EGFP* in a background with *Mst27D*⁺ function (*Mst27D*⁺) or in *Mst27D*ᶜᶜ/ *Df(2L)ade3* mutants (*Mst27D*⁻) were used for whole mount preparations that were labeled with a DNA stain. Single optical sections display regions from the clustered spermatid nuclei at high magnification. Scale bars = 5 μm.
(PDF)

**S1 Movie. Exit from M II followed by NE polarization in early round spermatids.** Time-lapse imaging was done with cysts released from dissected testes expressing EGFP-Nup358 (green) and Mst27D-mCherry (magenta). Movie starts at onset of anaphase II and displays maximum intensity projections of z-stacks acquired at 2 min intervals of a region initially containing a few secondary spermatocytes for the indicated time points (h:min:sec).
(MP4)

**S2 Movie. NPC-NE shedding in spermatids after nuclear elongation.** Time-lapse imaging was done with cysts released from dissected testes expressing Nup58-EGFP (white) and ProtB-DsRed (magenta). Movie displays maximum intensity projections of seven optical sections with 500 nm spacing from z-stacks acquired at 5 min intervals from a region containing a few elongated spermatid nuclei after drift correction with time indicated (h:min:sec).
(MP4)

**S3 Movie. Microtubule organization accompanying nuclear elongation in spermatids.** Time-lapse imaging was done with cysts released from dissected testes expressing EGFP-β-tubulin (white). Movie displays maximum intensity projections acquired at 1 min intervals, starting with 532 frames acquired from a cyst that exits from M II and proceeds until onset of cyst polarization. The subsequent 532 frames are from a cyst progressing through cyst polarization to early nuclear elongation and the final 532 frames are from a cyst completing nuclear elongation. Time (h:min:sec) is indicated.
(MP4)

**S4 Movie. Microtubule bundling in the dense complex around the start of nuclear elongation.** Time-lapse imaging was done with cysts released from dissected testes expressing EGFP-α-tubulin (green) and histone H2Av-mRFP (magenta). Movie displays maximum intensity projections of a region containing some of the clustered nuclei with time indicated (h:min:sec).
(MP4)

**S5 Movie. Microtubule bundling in the dense complex during progress of nuclear elongation.** Time-lapse imaging was done with cysts released from dissected testes expressing EGFP-β-tubulin (green) and histone H2Av-mRFP (magenta). Movie displays maximum intensity projections of a region containing some of the clustered nuclei with time indicated (h:min:sec).
(MP4)

**S6 Movie. Microtubule bundling in the dense complex during the final phase of nuclear elongation.** Time-lapse imaging was done with cysts released from dissected testes expressing EGFP-α-tubulin (green) and Mst27D-mCherry (magenta). Movie displays maximum intensity projections of a region containing some of the clustered nuclei with time indicated (h:min:sec).
(MP4)

**S7 Movie. Failure of microtubule bundling in the dense complex during nuclear elongation in *Mst27D* mutants.** Time-lapse imaging was done with cysts released from dissected testes expressing EGFP-α-tubulin (green) and histone H2Av-mRFP (magenta). Movie displays maximum intensity projections of a region containing some of the clustered nuclei with time indicated (h:min:sec).
(MP4)

**S1 Table. Source data.**
(XLSX)

**S2 Table. Sources of mutant alleles and transgenes.**
(PDF)

**S3 Table. Genotypes used in experiments.**
(PDF)

**S4 Table. Synthetic DNA sequences.**
(PDF)

## Acknowledgments

We thank Detlev Buttgereit, Renata Bastos, José C. Pastor-Pareja, and Bernhard Hampoelz for providing fly lines, Mary Dasso for antibodies against Nup358, and Bernd Roschitzki, Federico

Uliana, Fabian Frommelt, Joe Weber, Zeynep Kabakci and the Functional Genomics Center Zurich/UZH/ETHZ for the support of the AP-MS experiments. Moreover, we are grateful to Sina Moser and Hiro Yamada for their technical support.

## Author Contributions

**Conceptualization:** Christian F. Lehner.

**Data curation:** Pengfei Li, Giovanni Messina, Christian F. Lehner.

**Formal analysis:** Pengfei Li, Giovanni Messina, Christian F. Lehner.

**Funding acquisition:** Christian F. Lehner.

**Investigation:** Pengfei Li, Giovanni Messina, Christian F. Lehner.

**Project administration:** Christian F. Lehner.

**Supervision:** Christian F. Lehner.

**Validation:** Pengfei Li, Christian F. Lehner.

**Visualization:** Pengfei Li, Christian F. Lehner.

**Writing – original draft:** Christian F. Lehner.

**Writing – review & editing:** Pengfei Li, Giovanni Messina.

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
