## [Decision Letter · Decision Letter 0]

10 May 2023

Dear Dr Lehner,

Thank you very much for submitting your Research Article entitled 'Nuclear elongation during spermiogenesis depends on physical linkage of nuclear pore complexes to bundled microtubules by Drosophila Mst27D' to PLOS Genetics.

The manuscript was fully evaluated at the editorial level and by independent peer reviewers. The reviewers appreciated the attention to an important topic but identified some concerns both experimental and on the organization of the manuscript, that we ask you to address in a revised manuscript.

We therefore ask you to modify the manuscript according to the review recommendations. Your revisions should address the specific points made by each reviewer.

Yours sincerely,

Jean-René Huynh

Academic Editor

PLOS Genetics

Gregory P. Copenhaver

Editor-in-Chief

PLOS Genetics

Reviewer's Responses to Questions

**Comments to the Authors:**

Reviewer #1: i have uploaded my comments

Reviewer #2: This is a comprehensive paper by Li et al identifying Mst27D as the missing link between nuclear anchoring and microtubule attachment that helps bring about the dramatic nuclear lengthening that happens in Drosophila melanogaster spermiogenesis.Although the microtubule functioning link was hinted at by the previous discovery of EB1 homology domains, the identification of protein-protein interactions with nuclear pore components and the microtubules are convincing and very well described with a number of novel reagents. Although the quality of illustrations are low resolution in the reviewer version, it is clear that the cytological analyses presented here are also the most compelling descriptions of the nuclear elongation process. The logic of the arguments presented and the compelling genetic and cytological tools are exemplary and quite indicative of the detail oriented work of this group.

I do not have any major comments about the work. However, I was left a bit unsatisfied about the lack of statistical support about the degree of homology with EB1 (Figure 1 alignment) and the relative history of Mst27D. A previous paper suggested that this gene was a result of retroposition of CG15306, so it would have been useful to see at least some acknowledgement of the evolutionary relationships and possible retention of these genes in different Drosophila species, which should be very straightforward to do. The authors comment on how these additional EB1 homology genes might be slightly redundant for nuclear elongation function with Mst27D- it would have been really cool to see if double mutants of Mst27D and CG15306 are completely sterile due to redundancies in nuclear elongation function (whereas spag4 mutants are sterile). However, I do concede that is beyond the scope of the current paper (although the evolutionary description would be still nice here) which is still a very elegant piece of work.

Reviewer #3: The nuclei of developing Drosophila melanogaster sperm undergo a 200-fold reduction in volume, accompanied by extensive elongation to form needle-shaped sperm heads that are capable of penetrating the oocyte during fertilization. Although it has been known for ~50 years that nuclear elongation is accompanied by relocalization of nuclear pores and microtubules to a region called the dense complex, little progress has been made in understanding the mechanism by which this occurs. Here, the authors identify Mst27D, a Drosophila EB1 family member, as a molecular linker that ties the nuclear pore complex and nuclear envelope to centrosomal microtubules, thereby promoting formation of the dense complex and nuclear elongation. The manuscript addresses an important scientific question, provides a molecular mechanism for the link, and shows that this process is needed for male fertility. Overall, the experiments are elegantly done (including especially the time-lapse micrographs of dense body formation and MT bundling), the manuscript is logically organized and beautifully written, and the results will clearly advance the field. I have mostly minor comments to improve some aspects of the manuscript.

General comments:

1. The section of the Fig. 1B legend that describes relocalization of the NPC “into a part of the NE that covers only a hemisphere of the spherical nucleus in early spermatids” does not entirely reflect the relevant biology. Restriction of NPCs to a hemisphere of the spherical early spermatid nucleus actually occurs prior to the stage shown in the left-most diagram in 1B, and the basal body attaches in the middle of the hemisphere where the NPCs are located (see Galletta et al. 2020). After this, there is a 90°shift in the distribution of NPCs such that their distribution resembles what is shown in the left-most diagram. It might help to clarify this by mentioning the earlier step and saying it is not shown. This would avoid confusion on the part of readers who are familiar with the earlier step.

2. Three experiments that were performed using S2R+ cells could be extended to show whether the results are physiologically relevant in vivo: a) To confirm the interaction of Mst27D and Nup358, it would be nice to show coIP of the two proteins from testis extracts. b) It appears that mCherry-Nup358 localization at the NE is patchy when coexpressed with Mst27D_CH_EGFP in S2R+ cells. It would be nice to know whether expression of Mst27D_CH_EGFP also interferes with proper NE distribution of Nup358 in spermatocytes. c) Given the dramatic bundling of MTs caused by Mst27D overexpression in S2R+ cells, it would be interesting to know whether this also occurs upon Mst27D overexpression in the male germline.

3. The levels of Mst27D protein appear reduced following Nup358 knockdown (S2R+ cells) or degradation (testes). It would be helpful to include immunoblots showing the extent of this effect, especially in the testis degradFP experiments.

4. The authors state that Mst27D mutants have “partial individualization defects”. To support this claim, it would be helpful to include low-mag images of whole mount testes stained with fluorescent phalloidin, as this would indicate to what extent individualization complexes progress along the spermatid bundles in Mst27D mutants vs. controls.

5. Spag4 does not appear to be tightly juxtaposed to the nuclear envelope in the early elongating stage spermatids shown in S12B. Could a defect in basal body attachment precede the defects in nuclear elongation in Mst27D mutants?

6. The results section describing the difference in expression of transgenes with SV40 vs. endogenous 3’ regulatory sequences is rather long and could be shortened.

7. The student t-test requires that data be normally distributed. Given the distribution of datapoints in the graphs in Fig. 4D, S3C, and S8B, a Wilcoxon rank sum test might be more appropriate.

Comments on the figures and figure legends:

Fig. 3A: It would help to show the green and red single-channel images in grayscale and to change the red to magenta in the merged image to help colorblind readers.

Fig. 6C: The absence of a label under the right-most set of datapoints (shown as red dots) is confusing. It would help to indicate that these lack a transgene.

S7 Fig: The images in panel C appear to be 3D renderings, or at least qualitatively different from the images in the other panels. If true, this should be described in the legend.

S9 Fig: It would help to include labels indicating that the transgenes in panel A are expressed in a Mst28D+ background and those in panels B-D are in a Mst27D- background.

S10 Fig: It would help to include a label in panel B indicating that the sample was treated with colchicine.

Minor corrections:

line 62: pluralize “Histones”

line 73: change “role” to “roles”

line 74: unclear what is meant by “in context with”; perhaps “in the context of”?

line 91: missing “a” before “limited”

lines 102-103: for clarity, suggest moving “and the NPC-NE” (line 102) after “shaped nucleus” (line 103)

line 119: missing parenthesis after references

line 133: the abbreviation “g-“ is used throughout to indicate genomic upstream sequences, but this is never defined; it would help to define this here”

line 154: refer to Fig. 2A and 2B after “N terminus”

lines 155, 156, 160: replace “2B” with “2C” (three instances)

line 158: should this be “aa 151-424”, rather than “aa 141-424”, as suggested in Fig. 2A?

line 174: replace “2C” with “2B”

line 179: fix typo in “indicated”

line 186: delete “in” before figure callout

line 189: replace “highlighted” with “highlight”

lines 229-230: the results described here (line 229) are not present in S2 Fig (line 230)

lines 237-238: replace “S4 Fig” with “S3 Fig” (two instances)

line 267: missing “a” after “into”

line 272: the promoter “exumP” does not appear to be described anywhere in the text; please describe its origin, including what gene it is from, here

line 278: replace “5A” with “4B”

line 289: replace “5C” with “4D”

lines 290, 292,296: replace “5B,C” with “4C,D” (three instances)

lines 319, 332, 334: replace “Fig 6” with “Fig 5” (three instances)

line 383: replace “were” with “we”

line 392: replace “codes for” with “encodes”

line 409: replace “elongated” with “elongation”

line 463: delete “from”

lines 600-601: for clarity, suggest changing “appeared conceivable at first” to “initially appeared conceivable”

line 619: delete comma after “mutants”

lines 641, 650: awkward to use “according to our proposal” twice in two paragraphs; perhaps replace the first instance with “our proposed model” or “our proposed mechanism” and the second with “We propose that”

line 654: replace “be” with “by”

lines 713-718: paragraph is awkward as written; suggest moving “The MT bundles…[25,26].” (lines 717-718) before “While a single MT bundle…” (line 713)

line 747: replace “does” with “do”

line 780: replace “present” with “presence”

line 808: clarify whether the mutation described (N92T) is in the Mst27D coding region

lines 909-910: replace “I performed fertility tests” with “fertility tests were performed”

line 933: delete extra “d” after “inserted”

line 966: fix typo in “complete”

line 1013: delete reference (Lidsky et al. 2013) and include in reference list

lines 1211-1212: suggest deleting “we again subtracted” (line 1211) and inserting “were subtracted” before the period (line 1212)

line 1531: replace “pores” with “pore”

line 1558: pluralize “Positions”

line 1567: replace “was” with “were”

lines 1602-1603: for clarity, perhaps rearrange to say “with (Mst27D+) or without (Mst27D-) endogenous Mst27D expression using the indicated antibodies…”

line 1662: replace “of” with “or”

lines 1686-1689: could the effect on Mst27D levels be due to instability of the Mst27D protein following knockdown of its binding partner Nup358?

lines 1723, 1725: replace “case of” with “the” (two instances)

line 1771: replace “CH” with “CT”

lines 1775-1776: confusingly worded: are the images in panels B-D all in a Mst27D- background or are some of them in Mst27D+?

lines 1809-1816: the descriptions of panels A and B are switched compared to the order of the images shown

**Have all data underlying the figures and results presented in the manuscript been provided?**

Reviewer #1: Yes

Reviewer #2: Yes

Reviewer #3: Yes

PLOS authors have the option to publish the peer review history of their article (what does this mean?). If published, this will include your full peer review and any attached files.

Reviewer #1: No

Reviewer #2: **Yes: **Harmit Singh Malik

Reviewer #3: No

---

## [Editor Report · Decision Letter 1]

22 Jun 2023

Dear Dr Lehner,

We are pleased to inform you that your manuscript entitled "Nuclear elongation during spermiogenesis depends on physical linkage of nuclear pore complexes to bundled microtubules by Drosophila Mst27D" has been editorially accepted for publication in PLOS Genetics. Congratulations!

Yours sincerely,

Jean-René Huynh

Academic Editor

PLOS Genetics

Gregory P. Copenhaver

Editor-in-Chief

PLOS Genetics

Comments from the reviewers (if applicable):

**Data Deposition**

http://datadryad.org/submit?journalID=pgenetics&manu=PGENETICS-D-23-00368R1

**Press Queries**

---

## [Editor Report · Acceptance letter]

3 Jul 2023

PGENETICS-D-23-00368R1 

Nuclear elongation during spermiogenesis depends on physical linkage of nuclear pore complexes to bundled microtubules by Drosophila Mst27D 

Dear Dr Lehner, 

We are pleased to inform you that your manuscript entitled "Nuclear elongation during spermiogenesis depends on physical linkage of nuclear pore complexes to bundled microtubules by Drosophila Mst27D" has been formally accepted for publication in PLOS Genetics! Your manuscript is now with our production department and you will be notified of the publication date in due course.

With kind regards,

Anita Estes

PLOS Genetics

On behalf of:
